# UL2: Unifying Language Learning Paradigms

**Yi Tay⋆, Mostafa Dehghani⋆, Vinh Q. Tran, Xavier Garcia, Jason Wei, Xuezhi Wang,
Hyung Won Chung, Dara Bahri, Tal Schuster, Huaixiu Steven Zheng
Denny Zhou, Neil Houlsby, Donald Metzler**
Google Research, Brain Team
{yitay,dehghani}@google.com

## Abstract

Existing pre-trained models are generally geared towards a particular class of
problems. To date, there seems to be still no consensus on what the right architec-
ture and pre-training setup should be. This paper presents a unified framework for
pre-training models that are universally effective across datasets and setups. We
begin by disentangling architectural archetypes with pre-training objectives – two
concepts that are commonly conflated. Next, we present a generalized and uni-
fied perspective for self-supervision in NLP and show how different pre-training
objectives can be cast as one another and how interpolating between different
objectives can be effective. We then propose Mixture-of-Denoisers (MoD), a pre-
training objective that combines diverse pre-training paradigms together. We fur-
thermore introduce a notion of mode switching, wherein downstream fine-tuning
is associated with specific pre-training schemes. We conduct extensive ablative
experiments to compare multiple pre-training objectives and find that our method
pushes the Pareto-frontier by outperforming T5 and/or GPT-like models across
multiple diverse setups. Finally, by scaling our model up to 20B parameters, we
achieve SOTA performance on 50 well-established supervised NLP tasks rang-
ing from language generation (with automated and human evaluation), language
understanding, text classification, question answering, commonsense reasoning,
long text reasoning, structured knowledge grounding and information retrieval.
Our model also achieve strong results at in-context learning, outperforming 175B
GPT-3 on zero-shot SuperGLUE and tripling the performance of T5-XXL on one-
shot summarization. Finally, we show that UL2 20B works well with chain-of-
thought prompting and reasoning tasks, making it an appealing choice for research
into reasoning at a small to medium scale of 20B parameters. We publicly release
Flax-based T5X model checkpoints for the 20B model.

## 1 Introduction

*Note: This is a static copy of this paper as of the ICLR submission. Please use the arxiv version for
future updates : https://arxiv.org/abs/2205.05131. TYVM.*

There is a wide spectrum of pre-trained model options for NLP researchers and practitioners these
days (Devlin et al., 2018; Brown et al., 2020; Raffel et al., 2019; Radford et al., 2019; Liu et al.,
2019; Yang et al., 2019; Thoppilan et al., 2022; Fedus et al., 2021; Du et al., 2021; Chowdhery et al.,
2022). When faced with the question of what model should one use, the answer is often *it depends*,
followed by *on what task?*

Answering this can be overwhelming, comprising of a number of fine-grained follow-up questions
like, *'encoder-only or encoder-decoder?'*, *'span corruption or language model?'*. Pressing further,
the answer always seems to depend on the target downstream task. This paper questions and rethinks
this thought process, specifically answering the questions of *why should the choice of the pre-trained
LM depend on the downstream task?* and *how can we pre-train models that work universally well
across many tasks?*.

This paper proposes a step towards making a universally applicable language model possible. We
present a framework for *Unifying Language Learning Paradigms* or UL2 in short, that is consistently

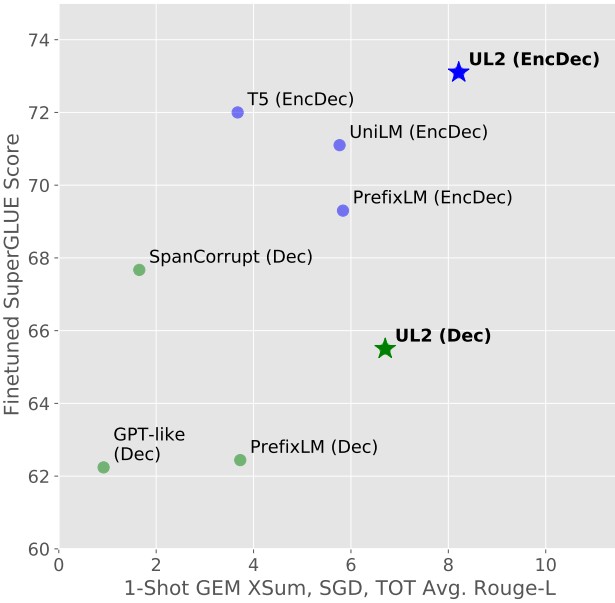

Figure 1: In both decoder-only and encoder-decoder setups, UL2 strikes a significantly improved balance in performance between fine-tuned discriminative tasks and prompt-based 1-shot open-ended text generation than previous methods. Note: Dec and EncDec are compute matched but EncDec models have double the parameters.

effective across a very diverse set of tasks and setups. Figure 1 shows an example of how UL2 can perform universally well, unlike other models that often have to make a trade-off.

The appeal of a universal model is clear, i.e., as this not only allows concentrated effort in improving and scaling a single model, instead of diversifying resources across $N$ models. Moreover, under resource constrained settings where only a few models can be served (e.g., on device), it would be preferable to have a single pretrained model that can perform well on many types of tasks.

At the core of UL2 is a the newly proposed Mixture-of-Denoisers (MoD), a pre-training objective that enables strong performance across tasks. MoD is a mixture of several well-established denoising objectives along with new ones; namely X-denoising (extreme denoising) which considers extreme span lengths and corruption rates, S-denoising (sequential denoising) that strictly follows sequence order, and R-denoising (regular denoising) that is a standard span corruption objective introduced in (Raffel et al., 2019). We show that MoD is conceptually simple but highly effective for a diverse set of tasks.

Our approach exploits the realization that most (if not all) well-studied pre-training objectives differ in the type of context a model is conditioned on. For example, the span corruption objective is akin to invoking multiple regions of prefix language modeling (PLM) (Liu et al., 2018; Raffel et al., 2019) whereby prefixes are contiguous segments of non-corrupted tokens and targets have full access to prefixes of all PLM segments. The setting where the span approaches the full sequence length is approximately a language modeling objective conditioned on long-range context. Thus, we are able to design a pre-training objective that smoothly interpolates these different paradigms (span corruption vs language modeling vs prefix language modeling).

It is also easy to see that each denoiser is difficult in different ways. They also differ in the nature of *extrapolation* (or interpolation). For example, bounding a model by bidirectional context (or the future) (ie.., span corruption) makes the task easier and becomes more akin to fact completion. Meanwhile, PrefixLM/LM objectives are generally more 'open ended'. These behaviours can be easily observed by monitoring the cross entropy losses of these different denoising objectives.

Given the MoD formulation, we conjecture that it is beneficial for our model to not only distinguish between different denoisers during pre-training but also to adaptively switch modes when learning

downstream tasks. We introduce mode switching, a new concept that associates pre-training tasks with dedicated sentinel tokens and allows dynamic mode switching via discrete prompting. Our model is able to switch modes between R,S and X denoisers on-demand after being pre-trained.

We then disentangle the architecture from the self-supervision scheme. While it might be a common misconception, as previously noted in Raffel et al. (2019), that a pre-trained model is strongly characterized by its backbone architecture (e.g., decoder-only vs. encoder-decoder), we find that the choice of the denoiser has significantly more impact. MoD supports either backbone, similar to how T5's span corruption may be trained with a decoder-only model. As such, UL2 is agnostic to architecture. We argue that the choice of backbone architecture is mainly a trade-off across different efficiency metrics.

We conduct systematic and ablative experiments on a suite of 9 diverse tasks aimed to capture different problem formulations (supervised and prompt-based in-context few-shot learning). We experiment with the SuperGLUE suite (Wang et al., 2019), and three tasks from the GEM benchmark (Gehrmann et al., 2021). In addition, we evaluate on open text generation, as well as prompt-based one-shot settings on all tasks. In this ablative setup, our experimental results show that UL2 outperforms T5 and GPT-like baselines on all 9 setups. On average, UL2 outperforms a T5 baseline by +43.6% and a language model by +76.1%. Among all the other competitive baselines considered, UL2 is the only method that outperforms T5 and GPT-like models on all tasks.

We scale UL2 up to a moderate scale setting of approximately 20B (19.5 to be exact) parameters. We publicly release T5X-based Flax checkpoints of the trained 20B UL2 model. We run experiments across a very diverse suite of 50+ NLP tasks ranging from language generation (with automated and human evaluation), language understanding, text classification, question answering, commonsense reasoning, long text reasoning, structured knowledge grounding and information retrieval. Our results show that UL2 achieves SOTA on a vast majority of tasks and setups.

Additionally, we conduct zero/few-shot experiments and show that UL2 outperforms GPT-3 175B on zero shot SuperGLUE. When compared with newer state-of-the-art models like GLaM (Du et al., 2021), PaLM (Chowdhery et al., 2022) and ST-MoE (Zoph et al., 2022), UL2 remains competitive at a compute-matched setup despite only training on C4 corpus which is known to be less effective than specially curated datasets used in (Du et al., 2021; Chowdhery et al., 2022). We delve into understanding trade-offs between zero-shot and finetuning performance and show that UL2 is Pareto-efficient with respect to both learning paradigms. On one-shot summarization, UL2 triples the performance of an LM adapted T5 XXL model and is competitive with (or outperforms) PaLM and LaMDA at the same compute cost.

Finally, we conduct experiments on reasoning tasks using the recent chain-of-thought (CoT) prompting (Wei et al., 2022b) and shows that UL2 20B benefits from CoT, making it an appealing and accessible open source option to conduct research on the reasoning capabilities language models.

## 2 UNIFYING LANGUAGE LEARNING PARADIGMS (UL2)

This section describes the UL2 framework and the proposed pre-training objectives which we study for the remainder of the paper.

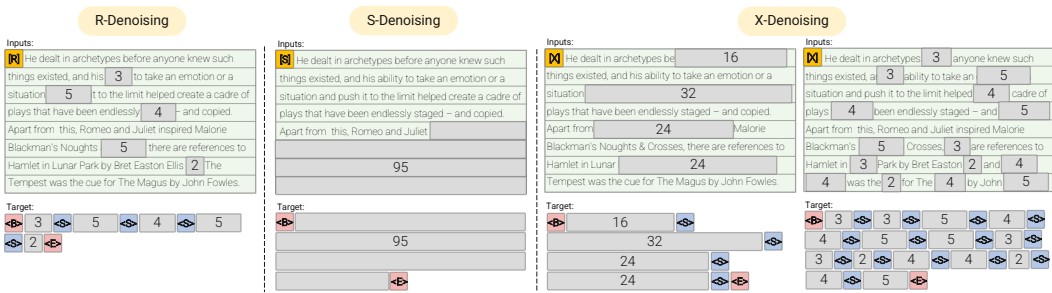

Figure 2: Mixture of denoisers for training UL2. Greyed out rectangles are masked tokens that are shifted to 'targets' for prediction.

## 2.1 Unified Perspective for Pre-training Tasks

Many pre-training tasks can be simply formulated as an 'input-to-target' task, wherein the input refers to any form of memory or context that the model conditions on, and the target is the model's expected output. Language models use all previous time-steps as inputs to the model to predict the next token, which is the target. In span corruption, the model leverages all uncorrupted tokens from the past *and* future as inputs for predicting the corrupted span (targets). Prefix-LMs are LMs that use past tokens as inputs, but consume the inputs bidirectionally: this offer more modelling power than unidirectional encoding of inputs in vanilla LM.

Given this perspective, we can approximately reduce one pre-training objective to another. For instance, in the span corruption objective, when the corrupted span, i.e., target, is equal to the entire sequence, the problem becomes effectively[1] a language modeling problem. With this in mind, using span corruption, by setting the span length to be large, we can effectively mimic the LM objective in local regions.

We define a notation that covers all of the different denoising tasks that we use in this paper. The inputs and targets of the denoising tasks are generated by a SPANCORRUPT function that is parameterized by three values $(\mu, r, n)$, where $\mu$ is the mean span length, $r$ is the corruption rate, and $n$ which is number of corrupted spans. Note that $n$ may be a function of the input length, $L$, and the span length $\mu$, e.g. $L\mu$, but in some cases, we use a fixed value of $n$. Given an input text, SPAN-CORRUPT introduces corruptions to the spans of lengths that are drawn from a (normal or uniform) distribution with mean of $\mu$. After corruption, the input text is then fed to the denoising task and the corrupted spans are used as targets to be recovered.

As an example, to construct an objective analogous to causal language modeling using this formulation, one would simply set $(\mu = L, r = 1.0, n = 1)$, i.e. a single span with its span length equal to the length of the sequence. To express one similar to Prefix LM objective, one would set $(\mu = L-P, r = 1.0 - P/L, n = 1)$ where $P$ is the length of the prefix, with the additional constraint that the single corrupted span always reaches the end of the sequence.

We note that this inputs-to-targets formulation can be applied to both encoder-decoder models and single-stack transformer models (e.g., decoder models). We opt to select models that predict the next target token instead of those that do so in-place (e.g., predict the current masked token in BERT) because the next-target formulation is more general and can subsume more tasks instead of using a special "`CLS`" tokens and task-specific projection heads.

## 2.2 Mixture of Denoisers

We conjecture that a strong universal model has to be exposed to solving diverse set of problems during pre-training. Given that pre-training is done using self-supervision, we argue that such diversity should be injected to the objective of the model, otherwise the model might suffer from lack a certain ability, like long-coherent text generation.

Motivated by this, as well as current class of objective functions, we define three main paradigms that are used during pre-training:

- **R-Denoiser** - The regular denoising is the standard span corruption introduced in Raffel et al. (2019) that uses a range of 2 to 5 tokens as the span length, which masks about $15\%$ of input tokens. These spans are short and potentially useful to acquire knowledge instead of learning to generate fluent text.

- **S-Denoiser** - A specific case of denoising where we observe a strict sequential order when framing the inputs-to-targets task, i.e., prefix language modeling. To do so, we simply partition the input sequence into two sub-sequences of tokens as context and target such that the targets do not rely on future information. This is unlike standard span corruption where there could be a target token with earlier position than a context token. Note that similar to the Prefix-LM setup, the context (prefix) retains a bidirectional receptive field. We note that S-Denoising with very short memory or no memory is in similar spirit to standard causal language modeling.

---

[1]This is roughly approximate since the model still conditions on a sentinel token.

- **X-Denoiser** - An extreme version of denoising where the model must recover a large part of the input, given a small to moderate part of it. This simulates a situation where a model needs to generate long target from a memory with relatively limited information. To do so, we opt to include examples with aggressive denoising where approximately $50\%$ of the input sequence is masked. This is by increasing the span length and/or corruption rate. We consider a pre-training task to be extreme if it has a long span (e.g., $\geq 12$ tokens) **or** have a large corruption rate (e.g., $\geq 30\%$). X-denoising is motivated by being an interpolation between regular span corruption and language model like objectives.

This set of denoisers has strong connections with previously used objective functions: R-Denoising is the T5 span corruption objective, S-Denoising is connected to causal language models that are GPT-like, and X-Denoising can expose the model to a combination of objectives from T5 and Causal LMs. Notably, X-denoisers are also connected to improve sample efficiency since more tokens are learned to be predicted in each sample, in similar spirit to LMs. We propose blending all these tasks in a uniform fashion and have a hybrid self-supervised objective. The final objective is a mixture of 7 denoisers that are configured as follows:

| Denoiser | Setting |
|----------|---------|
| R | $(\mu = 3, r = 0.15, n) \cup (\mu = 8, r = 0.15, n)$ |
| S | $(\mu = L4, r = 0.25, 1)$ |
| X | $(\mu = 3, r = 0.5, n) \cup (\mu = 8, r = 0.5, n) \cup (\mu = 64, r = 0.15, n) \cup (\mu = 64, r = 0.5, n)$ |

Table 1: Configuration of UL2's mixture-of-denoisers used in the paper.

For X- and R-Denoisers, the span length is sampled from a normal distribution with mean of $\mu$. For S-Denoisers, we use a uniform distribution, fix the number of corrupted spans to 1, and have an additional constraint that the corrupted span should end at the end of the original input text, i.e. no un-cropped token should appear after the corrupted part. This is roughly equivalent to seq2seq denoising or the Prefix LM pre-training objective.

Since LM is a special case of Prefix-LM, we did not find it necessary to include a casual LM task into the mixture. All tasks have an approximate equal participation in the mixture. We also explore an alternative where we increase number of S-denoisers up to $50\%$ of denoisers in the Mixture and all other denoisers take up the remainder. We present detailed ablation studies of various design choices in the later sections.

Finally, the mixing in Mixture-of-Denoisers is what makes it universally powerful. Alone, some of the denoiser types do not perform well. For instance, the original T5 paper explored an option with 50% corruption rate (X-denoising) and found that to not work well.

The implementation of UL2's mixture of denoiser is simple and easy to implement using a library like seqio[2] (Roberts et al., 2022). See appendix for more details on implementation.

## 2.3 MODE SWITCHING

We introduce the notion of paradigm-shifting via mode switching. During pre-training, we feed the model an extra *paradigm* token, i.e., {[R], [S], [X]} that helps the model switch gears and operate on a mode that is more suitable for the given task. For fine-tuning and downstream few-shot learning, to trigger the model to learn better solutions, we also add a paradigm token with respect to the setups and requirements of the downstream task. Mode switching in fact binds downstream behavior to one of the modes we used during upstream training.

## 2.4 MODEL ARCHITECTURE

UL2 adopts an architecture-agnostic philosophy. We argue that the choice between both architectures (encoder-decoder vs decoder-only) is a more of an efficiency trade-off and that architecture

---

[2]https://github.com/google/seqio

Table 2: Experimental results on a suite of language understanding and generation tasks on both supervised and one-shot setup. Models are pretrained on 32B tokens.

| Obj | Arch | Params | Supervised Finetuning | | | | In-context One-shot | | | | |
|-----|------|--------|-------|-------|-------|-------|-------|-------|-------|-------|-------|
| | | | SG | XS | SGD | TOT | SG | XS | SGD | TOT | LM |
| CLM | Dec | 167M | 62.24 | 28.18 | 55.44 | 59.40 | 39.22 | 1.16 | 1.40 | 0.20 | -2.35 |
| PLM | Dec | 167M | 62.44 | 28.21 | 55.55 | 59.52 | 42.54 | 1.08 | 3.70 | 6.40 | -2.54 |
| SC | Dec | 167M | 67.67 | 29.14 | 55.48 | 60.47 | 38.53 | 1.16 | 2.20 | 1.60 | -3.62 |
| SCLM | Dec | 167M | 63.36 | 29.02 | 55.71 | 60.00 | 40.78 | 3.03 | 1.27 | 0.10 | -2.38 |
| UL2 | Dec | 167M | 65.50 | 28.90 | 55.80 | 60.39 | **42.30** | 8.01 | 6.30 | 5.80 | **-2.34** |
| PLM | ED | 335M | 69.30 | **31.95** | 55.70 | 60.91 | 38.18 | 6.50 | 7.11 | 3.90 | -2.42 |
| SC | ED | 335M | 72.00 | 31.05 | 55.80 | 61.25 | 38.51 | 7.49 | 1.43 | 2.10 | -7.23 |
| SCLM | ED | 335M | 72.50 | 31.69 | 55.70 | 60.94 | 39.74 | 5.13 | **8.70** | **7.30** | -2.40 |
| UniLM | ED | 335M | 71.10 | 31.00 | 55.83 | 61.03 | 39.86 | 6.70 | 6.50 | 4.10 | -2.65 |
| UL2 | ED | 335M | **73.10** | 31.86 | **56.10** | **61.50** | 41.30 | **11.51** | 6.63 | 6.50 | -2.55 |

choice should not be conflated with the pretraining objective. Hence, we have both a UL2 decoder and UL2 encoder-decoder in similar spirit to how there are multiple sizes per model. We discuss this efficiency trade-off in detail in our experiment section. UL2 adopts a pretty standard vanilla T5 Transformer that have been enhanced with modifications that have withstood the test of time, i.e., GLU layers (Shazeer, 2020) and T5-style relative attention. To not further conflate architectural modifications with pretraining contributions, the backbone of the model remains similar to a T5-like model. This is also in light of results such as (Narang et al., 2021).

## 3 ABLATIVE EXPERIMENTS

This section describes our ablative experimental setup (e.g., baselines, datasets, implementation details) and results. Our overall findings show that UL2 outperforms T5-like and GPT-like models on 9 out of 9 tasks.

**Baselines** For pre-training objectives, we compare with the following pre-training baselines: (1) Causal Language Model, (2) Prefix Language model, (3) Span Corruption (SC), (4) Span Corruption + LM (SCLM), and (5) UniLM (Dong et al., 2019). For all objectives, we explore both single-stack and encoder-decoder architectures. All architectures are inputs-to-targets either implemented in encoder-decoder or decoder-only model structures since we consider BERT-style masked language modeling pretraining to have already been effectively subsumed by this style of pretraining, as empirically made evident in (Raffel et al., 2019). Task-specific classification heads are also not recommended, since they clearly go against the principle of having a universal model (and are also very cumbersome).

**Experimental Setup** We conduct our experiments on a diverse set of supervised and prompt-based few-shot learning tasks. To enable the comparison between models from this perspective, we need an aggregate performance score. However, metrics on different tasks we include are widely different in nature – take, for example, F1 and perplexity. To address this, we opt to report and use the *normalized relative gain with respect to baselines* as an overall metric. For this purpose, we use the standard language model (decoder-only) (GPT-like) and standard span denoising encoder-decoder (T5) as prime baselines and report all methods against their relative performance against these well-established candidates. We believe this is the most suitable method for comparing these models since it is easy to reason about how much a new model is generally better than a popular setting (e.g., GPT or T5-like). The full details of our experimental setup, including metrics, datasets and implementtion details can be found in the Appendix.

### 3.1 OVERVIEW OF ABLATIVE EXPERIMENTAL RESULTS

Table 2 reports the raw results on all the benchmark tasks and datasets. To facilitate easier comparison across setups, we also report relative comparisons against well-established baselines such as T5 and GPT models. This is reported in Tables 3 and 4 respectively.

Table 3: Relative performance compared to standard encoder-decoder span corruption model (T5). Results in this table are expressed in terms of relative percentage improvements over a baseline. Model with ⋆ denotes the main compared baseline. Overall score column is normalized to be weighted equally across tasks.

| Obj | Arch | Supervised | | | | One-shot | | | | | | |
| | | SG | XS | SGD | TOT | SGL | XS | SGD | TOT | LM | All | Win |
| --- | --- | --- | --- | --- | --- | --- | --- | --- | --- | --- | --- | --- |
| CLM | Dec | -13.6 | -9.2 | -0.7 | -3.0 | +1.8 | -91.7 | -2.2 | -90.5 | +208 | -31.7 | 2/9 |
| PLM | Dec | -13.3 | -9.2 | -0.5 | -2.8 | +10.5 | -85.6 | +158 | +205 | +185 | -11.0 | 4/9 |
| SC | Dec | -5.6 | -6.2 | -0.6 | -1.3 | +0.05 | -84.5 | +54 | -23.8 | +99 | -20.6 | 3/9 |
| SCLM | Dec | -6.0 | -6.5 | -0.2 | -2.0 | +5.9 | -59.6 | -11.3 | -95 | +204 | -16.1 | 2/9 |
| UniLM | Dec | -10.1 | -8.2 | -0.2 | -2.3 | -5.3 | -69.1 | +382 | +110 | +200 | -16.1 | 3/9 |
| UL2 | Dec | -9.0 | -6.9 | 0.0 | -1.4 | +9.8 | +6.9 | +340 | +176 | +209 | **+14.1** | 5/9 |
| PLM | ED | -3.7 | +2.9 | -0.2 | -0.6 | -0.86 | -13.3 | +397 | +86 | +199 | +16.7 | 5/9 |
| SC⋆ | ED | 0.0 | 0.0 | 0.0 | 0.0 | 0.0 | 0.0 | 0.0 | 0.0 | 0.0 | 0.0 | - |
| SCLM | ED | +0.7 | +2.1 | -0.2 | -0.5 | +3.2 | -31.6 | +508 | +248 | +201 | +28.3 | 7/9 |
| UniLM | ED | -1.2 | -0.2 | +0.1 | -0.4 | +3.5 | -11.0 | +355 | +95 | +173 | +19.8 | 5/9 |
| UL2 | ED | +1.5 | +2.6 | +0.5 | +0.4 | +7.2 | +53.6 | +363 | +210 | +184 | **+43.6** | **9/9** |

Table 4: Relative performance compared to standard decoder causal language model (GPT-like). Results in this table are expressed in terms of relative percentage improvements over a baseline. Model with ⋆ denotes the main compared baseline. Overall score column is normalized to be weighted equally across tasks.

| Obj | Arch | Supervised | | | | One-shot | | | | | | |
| | | SG | XS | SGD | TOT | SG | XS | SGD | TOT | LM | All | Win |
| --- | --- | --- | --- | --- | --- | --- | --- | --- | --- | --- | --- | --- |
| CLM⋆ | Dec | 0.0 | 0.0 | 0.0 | 0.0 | 0.0 | 0.0 | 0.0 | 0.0 | 0.0 | 0.0 | - |
| PLM | Dec | +0.3 | +0.1 | +0.2 | +0.2 | +8.5 | +74.3 | +164 | +3100 | -8.0 | +21.4 | 8/9 |
| UniLM | Dec | +4.0 | +1.1 | +0.5 | +0.7 | -7.0 | +274 | +393 | +2100 | -2.5 | +21.0 | 7/9 |
| SC | Dec | +8.7 | +3.4 | +0.1 | +1.8 | -1.8 | +87.0 | +57.1 | +700 | -54.2 | +13.9 | 7/9 |
| SCLM | Dec | +1.8 | +3.0 | +0.5 | +1.0 | +4.0 | +387 | -9.3 | -50 | -1.3 | +15.8 | 6/9 |
| UL2 | Dec | +5.2 | +2.6 | +0.6 | +1.7 | +7.9 | +1190 | +350 | +2800 | +0.3 | **+45.7** | **9/9** |
| PLM | ED | +11.3 | +13.4 | +0.5 | +2.5 | -2.6 | +946 | +408 | +1850 | -2.9 | +48.6 | 7/9 |
| SC | ED | +16.5 | +10.2 | +0.6 | +3.1 | -1.8 | +1107 | +2.3 | +950 | -208 | +31.7 | 7/9 |
| SCLM | ED | +15.7 | +12.5 | +0.5 | +2.6 | +1.3 | +726 | +522 | +3550 | -2.2 | +60.3 | 8/9 |
| UniLM | ED | +14.2 | +10.0 | +0.7 | +2.7 | +1.6 | +974 | +365 | +1950 | -12.9 | +52.6 | 8/9 |
| UL2 | ED | +17.4 | +13.1 | +1.2 | +3.5 | +5.3 | +1754 | +373 | +3150 | -8.3 | **+76.1** | 8/9 |

**Decoder Vs Encoder-Decoder**   Before we dive into the results of this segment, we would like to remind readers that there is no easy way to compare decoder-only models with encoder-decoder models. In short, we can either compare them in a compute-matched setup or a parameter-matched way. Therefore, the encoder-decoder models in these set of results have approximately twice the number of parameters as the decoder models but have similar speeds.

We note that this may slightly favor encoder-decoders since this can be interpreted form of model sparsity. Moving back to the results, when using T5 as the reference baseline, we note that, with the exception of UL2 Decoder, none of the pre-trained decoders models outperform T5. Additionally, there is a 10% to 30% degradation in overall relative performance. The best decoder baseline model here is the Prefix-LM decoder model, which is about 10% worse than the T5 baseline. It is clear from these results that encoder-decoder models should be preferred over decoder-only models if and only if there is no concern about storage, i.e., parameter counts are generally less important than actual throughput (see (Dehghani et al., 2021a) for a detailed discussion).

When there is a parameter constraint, the Prefix-LM decoder makes for a suitable alternative. Finally, an interesting data point is how we were able to push the UL2 decoder to outperform the T5 encoder-decoder setup by +14.6%. That said, this UL2 decoder does not outperform our UL2 encoder-decoder. However, this reinforces our point that the self-supervision objective may be intrinsically more important than the backbone architecture and negotiating architectural choices is mainly about efficiency trade-offs that can be studied independently.

**Is GPT and/or T5 the optimal setup?**    Based on the relative comparisons against a GPT-like (causal LM + decoder) and T5-like (span corruption + encoder decoder) setup, we are able to easily identify if the well-established setups are indeed optimal or already close to optimal. Firstly, the causal LM (GPT-like) setup appears to be the worse configuration as it is outperformed by all our baselines. We thus make the straightforward recommendation of always at least training with Prefix-LM or UniLM whenever possible. The best decoder-only model (with the exception of UL2) is the Prefix-LM pre-training that keeps a memory prefix for a language model to condition on. Regarding Prefix-LM pre-training, it is interesting that Prefix-LM actually outperforms the T5 span corrupt setup by $+16.7\%$. The Prefix-LM encoder-decoder model is indeed less effective than the default T5 model on SuperGLUE but is on a whole, stronger especially when it comes to one-shot or open text-generation. Overall, between the Prefix-LM and the span corruption encoder-decoder model (T5), it is unclear to which is the universally superior model as there are gives and takes across the different sub-tasks although it is worthy noting the Prefix-LM EncDec model only sacrifices a minor degradation in certain tasks for a huge multifold increase in other tasks.

**On the Performance of UniLM and SCLM**    On the encoder-decoder setup, both the UniLM and SCLM objective performs better than the standard span corruption objective in terms of aggregated and normalized overall gain. This shows that, in general, mixing pre-training objectives is helpful. On the decoder setup, there is an overall gain of $+9.4\%$ gain for UniLM and $+16.1\%$ for SCLM compared to the baseline causal LM. In terms of individual tasks, UniLM and SCLM both outperforms T5 on 6 out of 9 tasks. It is also noteworthy that SCLM performs the best out of all models on 1shot generation (SGD and TOTTO).

**On the Performance of the Proposed UL2**    Finally, we note that UL2 performs the best when compared against both the GPT-like model and the T5-like model. Overall, UL2 outperforms by T5 $+43.4\%$ and $+76.2\%$ when compared to the GPT-like CLM decoder model. This is the highest relative (overall) gain compared to all other alternatives. We also note that on all individual tasks, UL2 outperforms T5 on **all 9 out of 9** considered tasks. Hence, UL2 is a universally better option compared to the span corruption T5 model. While UL2 doesn't always outperform all baselines on all individual tasks, UL2 is very consistent. Even when it loses to another method on a task, the loss is relatively marginal (e.g., 6.5 vs 7.3 on one-shot TOTTO). Conversely, when UL2 outperforms a baseline like T5, the gain can be as large as $+363\%$. UL2 remains the most consistently strong method. The consistent improvement also suggests that it can be used as a more consistent replacement to T5 and GPT-like models.

# 4    SCALING TO 20B PARAMETERS

We are also interested to evaluate UL2 in a scaled up setting. We trained a 20B UL2 model that we eventually released publicly. This section describes the highlights of our experimental results for UL2 20B.

## 4.0.1    EXPERIMENTS ON SUPERVISED FINETUNING

We conduct experiments over 50 NLP tasks under the supervised finetuning paradigm. These tasks range from language generation (with automated and human evaluation), language understanding, text classification, question answering, commonsense reasoning, long text reasoning, structured knowledge grounding and information retrieval. The full list of 50 tasks and their corresponding experimental results can be found in the Appendix in 11. Overall, our experimental results show that UL2 achieves state-of-the-art performance on around 50+ NLP tasks and setups. Finally, we note that UL2 20B does pretty well on human evaluation on GENIE tasks, outperforming sota on several metrics. This ascertains that the generation quality of UL2 is reasonably solid.

## 4.0.2    RESULTS ON ZERO-SHOT AND ONE-SHOT LEARNING

Table 5 reports results on zero-shot SuperGLUE. We note that UL2 20B outperforms GPT-3 and other compute-matched models on zero-shot NLU.

Table 5: Results on zero-shot learning on SuperGLUE dataset. We compare with GPT-3, GLaM and PaLM (Chowdhery et al., 2022). We also include models that are relatively compute-matched with UL20B such as T5-XXL with LM adaptation (Lester et al., 2021), GPT-3 13B and GLaM-8B dense. Notably, UL20B outperforms GPT-3 175B and all other models in a similar compute class on average score.

| Model | BoolQ | CB | RTE | ReCORD | WSC | WiC | COPA | MultiRC | Avg |
|---|---|---|---|---|---|---|---|---|---|
| ST-MoE-32B (269B) | 40.8 | 41.1 | 52.7 | 50.0 | 57.5 | 50.0 | 56.0 | 30.3 | 47.6 |
| GPT-3 175B | 60.5 | 46.4 | 63.5 | 90.2 | 65.4 | 0.0 | 91.0 | 72.9 | 61.2 |
| GLaM-MoE 1.2T | 83.0 | 33.9 | 68.8 | 90.3 | 84.9 | 50.5 | 90.0 | 45.1 | 68.3 |
| PaLM 540B | 88.0 | 51.8 | 72.9 | 92.9 | 89.1 | 59.1 | 93.0 | 83.5 | 78.8 |
| T5-XXL | 44.3 | 37.5 | 48.8 | 85.8 | 59.3 | 50.9 | 70.0 | 23.0 | 52.5 |
| GPT-3 13B | 66.2 | 19.6 | 62.8 | 89.0 | 64.4 | 0.0 | 84.0 | 71.4 | 57.2 |
| PaLM-Dense 8B | 68.3 | 41.1 | 54.2 | 87.8 | 78.9 | 47.0 | 86.0 | 47.5 | 63.9 |
| UL2 20B (*single ckpt*) | 63.1 | 41.1 | 60.7 | 88.1 | 79.9 | 49.8 | 85.0 | 36.2 | 63.0 |
| UL2 20B (*best*) | 63.1 | 50.0 | 60.7 | 88.1 | 80.6 | 55.2 | 88.0 | 36.2 | 65.2 |

Table 6: Results on One-Shot Summarization on XSUM.

| Model | Rouge-1 | Rouge-2 | Rouge-L |
|---|---|---|---|
| LaMDA 137B | - | 5.4 | - |
| PaLM 62B | - | 11.2 | - |
| PaLM 540B | - | **12.2** | - |
| PaLM 8B | - | 7.9 | - |
| T5 XXL 11B | 0.6 | 0.1 | 0.6 |
| T5 XXL 11B + LM | 13.3 | 2.3 | 10.7 |
| UL2 20B | **25.5** | **8.6** | **19.8** |

Table 14 reports results on 1-shot summarization. Our results show that the performance of UL2 20B is about 3x the performance of LM adapted T5 XXL model. Moreover, UL2 20B outperform LaMDA 137B and has better performance compared to PaLM 8B which is approximately compute-matched with UL2. The best result, however, is still the larger 540B and 62B PaLM models.

### 4.0.3 RESULTS ON CHAIN-OF-THOUGHT REASONING

We conduct experiments (full details in Appendix) to show that UL2 20B is capable of performing chain-of-thought reasoning (Wei et al., 2022b). Whereas most prior success on chain-of-thought has been shown on large and non-public models, UL2 is a comparatively smaller and publicly available.

Table 7: Chain-of-thought prompting and self-consistency (SC) results on five arithmetic reasoning benchmarks. Full details in Appendix.

| Model | GSM8K | SVAMP | ASDiv | AQuA | MAWPS | Average |
|---|---|---|---|---|---|---|
| UL2 20B: standard prompting | 4.1 | 10.1 | 16.0 | 20.5 | 16.6 | 13.5 |
| UL2 20B: CoT prompting | 4.4 | 12.5 | 16.9 | 23.6 | 19.1 | 15.3 |
| UL2 20B: CoT prompting + calc. | 6.9 | 28.3 | 34.3 | 23.6 | 42.7 | 27.2 |
| UL2 20B: CoT prompting + calc. + SC | **10.2** | **41.4** | **43.5** | **26.9** | **57.9** | **36.0** |

## 5 CONCLUSION

We proposed a new paradigm for training universally effective models. UL2 is characterized by two key ideas. Firstly, we propose a new Mixture of Denoisers (MoD) pretraining that frames multiple pretraining tasks as span corruption, diversifies and then mixes them. Secondly, we introduce mode switching, a way of associating downstream task behaviour to upstream pretraining. Extensive ablative experiments show that UL2 consistently outperforms GPT-like and T5 models on a wide range of supervised and few-shot tasks, outperforming T5 on 9 out of 9 tasks and by a normalized overall gain of +76.1%. Finally, we scale UL2 up to 20B parameters and conduct experiments on a diverse suite of 50 to 60 NLP tasks and setups. UL2 achieves sota performance on 50 of them. Pretrained checkpoints will be released at `https://github.com/anonymous`.

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

## A  EXPERIMENTAL SETUP FOR ABLATIVE EXPERIMENTS

### A.0.1  DATASETS AND TASKS

The datasets we use are SuperGLUE (Wang et al., 2019), comprising of 8 NLU sub-tasks. We also conduct experiments on 3 datasets from the GEM benchmark (Gehrmann et al., 2021) that focuses on language generation problems. We arbitrarily select XSUM (summarization), ToTTo (table-to-text generation) (Parikh et al., 2020) and Schema Guided Dialog (SGD) (Rastogi et al., 2019) from the GEM benchmark. For all these tasks, we evaluate on both supervised fine-tuning and prompt-based one-shot learning. Finally we also compare our models on their general ability for text generation using perplexity scores on the C4 validation set. We believe our suite of tasks gives us good coverage across many setups in the literature including supervised and conditional few-shot learning.

### A.0.2  METRICS AND HOLISTIC EVALUATION

For SuperGLUE, we report well-established metrics such as accuracy, F1 or Exact Match, whenever appropriate. For GEM benchmark, we use the Rouge-L metric. For language modeling we report negative log perplexity. The *universality* of the models, i.e., their collective performance across all range of tasks, is a main evaluation criteria here. To enable the comparison between models from this perspective, we need an aggregate performance score. However, metrics on different tasks we include are widely different in nature – take, for example, F1 and perplexity. To address this, we opt to report and use the *normalized relative gain with respect to baselines* as an overall metric. For this purpose, we use the standard language model (decoder-only) (GPT-like) and standard span denoising encoder-decoder (T5) as prime baselines and report all methods against their relative performance against these well-established candidates. We believe this is the most suitable method for comparing these models since it is easy to reason about how much a new model is generally better than a popular setting (e.g., GPT or T5-like). We also highlight the fact that the overall gain is **normalized**, so this becomes harder to exploit or be susceptible to benchmark lottery effects (Dehghani et al., 2021b).

### A.0.3  IMPLEMENTATION DETAILS

Our experiments are all conducted in JAX/Flax (Bradbury et al., 2018) using the open source T5X[3] framework (Roberts et al., 2022) and Flaxformer[4]. We pre-train all models for 500K steps with a batch size of 128 and a sequence length of 512 inputs and 512 targets using the C4 corpus. The total approximate tokens seen during pre-training is approximately 32 billion tokens. Each pre-training run is typically trained using 64 to 128 TPUv4 chips (Jouppi et al., 2020). We optimize our model with the Adafactor (Shazeer & Stern, 2018) optimizer with an inverse square root learning rate. To understand the trade-off of different backbone architectures, we run all baseline pre-training objectives with both the decoder-only architecture and encoder-decoder architecture. We report key experiment results using a base architecture of approximately 167M parameters for the decoder model and 335M parameters for the encoder-decoder model. All models use a standard Transformer that uses SwiGLU layers as described in (Shazeer, 2020). We utilize the default T5 English 32K sentencepiece for all models. Within the context of decoder-only models, except for the case of the decoder model trained on causal LM, our experiments always use a bidirectional receptive field **only** in it's input segment and autoregressive decoding at the *targets* segment. This is essentially the a PrefixLM-type architecture[5] (Raffel et al., 2019) which we find to be consistently better than a full causal decoder model.

### A.1  MODE SWITCHING ABLATIONS

In order to ascertain that our mode switching capabilities have an effective on performance, we conduct ablation experiments.

---

[3]https://github.com/google-research/t5x.
[4]https://github.com/google/flaxformer
[5]Not to be confused with the PrefixLM pretraining objective.

Table 9: Ablation study for Mixture-of-Denoisers. Span, Rate and SD are in percentages (%). We report SuperGLUE score (SG) and XSUM Rouge-L (XS).

| Name | Ablation Method | | | Supervised | | One-shot | |
|------|---------|----------|------|------|------|------|------|
| | Span ($\mu$) | Rate ($r$) | SD% | SG | XS | SG | XS |
| A | - | - | 100 | 69.3 | 31.1 | 38.2 | 6.5 |
| B | 3 | 50 | 0 | 72.0 | 32.0 | 38.5 | 7.5 |
| C | 3,8,12 | 15,50 | 14 | 71.9 | 32.1 | 38.6 | 4.1 |
| D | 3,8,12,32 | 15,50 | 11 | 71.0 | **32.2** | **42.7** | 10.6 |
| E | 3,8,32,64 | 15,50 | 11 | 73.1 | **32.2** | 40.7 | 10.4 |
| F | 3,8,64 | 15,50 | 17 | 70.6 | 31.6 | 41.3 | 11.5 |
| G | 3,8,32,64 | 15 | 25 | 69.2 | 31.6 | 42.4 | 10.1 |
| H | 8, 64 | 15 | 25 | 72.5 | 31.2 | 39.2 | 10.9 |
| I | 3,8,12, 32 | 15,50 | 50 | 71.2 | 32.0 | 38.1 | 11.7 |
| J | 3,8,64 | 15,50 | 50 | 71.3 | 31.6 | 38.1 | **11.8** |
| K | 3,8,12 | 15,50 | 0 | **73.7** | 32.0 | 39.3 | 2.6 |
| L | 3,8,64 | 15,50 | 0 | 70.1 | 32.1 | 38.0 | 7.3 |

Table 8: Effect of different paradigm prompts on 1-shot evaluation, using a Encoder-Decoder architecture pre-trained using UL2 on 7B tokens.

| Model/Prompt | 1Shot XSum | 1Shot SuperGLUE |
|--------------|------------|-----------------|
| Baseline T5 | 6.9/0.6/6.1 | 33.9 |
| UL2 / None | 13.2/1.4/10.8 | 38.3 |
| UL2 / [R] | **13.5/1.5/11.1** | 38.5 |
| UL2 / [S] | 11.6/1.2/10.0 | 38.5 |
| UL2 / [X] | 8.9/0.9/7.6 | **38.7** |

We conduct experiments on one-shot XSum and one-shot SuperGLUE. Table 8 reports the result of varying the paradigm prompt to the model. Firstly, we observe that the prompt has quite substantial effect on model performance – i.e., using the right or wrong prompt can lead to a 48% gap in performance (on XSum, Rouge-1). SuperGLUE, on the other hand, is less sensitive to prompting. On SuperGLUE, using prompts are almost always better than not using prompts during one-shot eval. However, for XSum, getting the prompt right seems to be crucial for good performance.

## A.2 MIXTURE-OF-DENOISERS ABLATIONS

We conduct extensive experiments to verify the effectiveness of individual objectives within the MoD objective. Table 9 reports results for these ablations. We report results for varying the mean span, and corruption rate, along with the percentage of S-denoising used (denoted by % SD)). Note that the total number of denoisers in a mixture is $\|Span\| \times \|Corrupt\_Rate\| + 1$. We label these configurations from Var-A through Var-J to refer to them easily.

**X-Denoising is Complementarily Effective but Does Not Suffice as a Standalone**     We observe that mixing Extreme Denoising is effective. Most of the best results across the board come from mixtures with long spans (e.g., 32 or 64). When compared with variants without long spans (Var-D vs. Var-C), we see that Var-D is strictly better. We also draw the readers attention to Var-H, which is a variant that only employs long spans. In general, Var-H performs poorly, suggesting that extreme denoising complements regular denoising but does not suffice in isolation. This also corroborates the result from Raffel et al. (2019) that shows that a 50% corruption rate does not perform well. This slightly conflicts with the finding of (Wettig et al., 2022) although our architectures use a inputs-to-targets form of pretraining instead of BERT-style masked language modeling.

**Small Amounts of S-Denoisers is Preferred**     We explore a setting where we scale S-denoisers to 50% of the entire MoD mixture. We find that this generally hurts performance. Hence, we make a conclusion that S-denoisers are necessary but only small amounts of S-denoisers ($\approx 20\%$)

are preferred. Var-K and Var-L also explore the case where there is no S-denoising at all. While performance on one task substantially improves (SuperGLUE), another substantially degrades (one-shot XSUM). Meanwhile for Var-L which is identical to var-F (but without S-denoising), performs on a whole, substantially worse. Hence, we showed that S-denoising is crucial.

## A.3 Modestly Scaling Model Size and Pretraining Data

We conduct additional experiments by scaling up both 1) the model size and 2) pre-training dataset size. Concretely, we scale the UL2 Encoder-Decoder model up to approximately 1B parameters and increase the number of pre-training tokens to 0.5 trillion tokens. Our motivation is to conduct a sanity check that the proposed formulation also works at different model scales and to observe if there are differences and implications at operating at a larger scale. Moreover, it has also become a staple for language model research to derive scaling laws (Kaplan et al., 2020; Tay et al., 2021b). Table 10 reports results in this scaled setting. At large scale, we find that the proposed of the UL2 encoder-decoder model is still competitive. A key difference now is that UL2 drops the SuperGLUE suite against T5 (1B). However, this is compensated by not only out-performing on 7 out of 8 tasks but also improving performance by 2-4 times on one-shot evaluation. The gains on supervised fine-tuning is smaller, but still noticeable across the board on XSUM, SGD and TOT.

Table 10: Experiments with moderately scaled up models in terms of model compute (e.g., 1B for EncDec and 0.5B for decoder-only) and dataset size (0.5T tokens).

| Model | SG | Finetuning | | | In-context Learning | | | |
| | | XS | SGD | TOT | SG | XS | SGD | TOT |
|---|---|---|---|---|---|---|---|---|
| GPT-like | 62.3 | 37.1/15.7/30.2 | 56.0 | 60.3 | 36.4 | 1.2/0.1/1.1 | 3.5 | 0.0 |
| T5 | **84.7** | 43.0/20.8/35.6 | 56.0 | 62.1 | 29.4 | 8.9/0.8/7.8 | 2.1 | 1.4 |
| UL2 | 83.3 | **43.3/21.0/35.9** | **56.5** | **62.6** | **45.4** | **15.4/2.5/11.1** | **9.6** | **7.8** |

## B 20B UL2 experimental setup

This section describes the setup for our 20B experiments.

### B.1 Pretraining and Model Configuration

We follow the same training protocol in earlier experiments by pretraining on the C4 corpus but by also scaling the number of tokens the model sees during pretraining. We use a batch size of 1024 and 512 TPUv4 chips for pretraining this model. The model is trained on a total of 1 trillion tokens on C4 (2 million steps). The sequence length is set to $512/512$ for inputs and targets. Dropout is set to 0 during pretraining. Pre-training took approximately slight more than one month for about 1 trillion tokens. We use the same mixture of denoisers as earlier sections. The model has 32 encoder layers and 32 decoder layers, $d_{model}$ of 4096 and $d_{ff}$ of 16384. The dimension of each head is 256 for a total of 16 heads. Our model uses a model parallelism of 8. We retain the same sentencepiece tokenizer as T5 of 32k vocab size. Hence, UL20B can be interpreted as a model that is quite similar to T5 but trained with a different objective and slightly different scaling knobs. Similar to earlier experiments, UL20B is trained with Jax and T5X infrastructure. We release and open source T5X-based model checkpoints of this 20B model.

### B.1.1 Setup and Implementation Details

We conduct experiments on both finetuning and in-context learning. For supervised finetuning, our models are continuously finetuned after $N$ pretraining steps where $N$ is typically from $50k$ to $100k$. In other words, after each $N$k steps of pretraining, we finetune on each downstream task and record its results. This is generally done in a manual fashion. While some tasks were finetuned on earlier pretrained checkpoints as the model was still pretraining, many were finetuned on checkpoints nearer to convergence that we release. As we continiously finetune, we stop finetuning on a task once it has reached *sota* to save compute. Finetuning is generally done in a per-task basis and not co-trained.

Details of tasks where co-training is performed is found in the appendix. We leave the combination of massive multi-task training (Aribandi et al., 2021) and UL2 to future work.

For supervised finetuning, we generally adopt a learning rate in the range of $\{5 \times 10^{-5}, 1 \times 10^{-5} \, 1 \times 10^{-4}\}$ using the Adafactor optimizer . The general recipe is that we reset Adafactor optimizer states and/or adopt a loss normalization based on the number of real target tokens. This is reminiscent of the PaLM finetuning setup (Chowdhery et al., 2022). Batch size is generally a range of 32 to 128 although we did not find much impact of batch size on finetuning performance. Many of the evaluated tasks were not tuned much and we only ran once or twice before performing leaderboard submissions.

### B.1.2 DATASETS FOR SUPERVISED FINETUNING

To demonstrate the universality of the approach, we consider a total of nearly 50+ NLP tasks. We list our categorization of tasks below. Note that the categorization of tasks are generally soft in nature and some tasks may cross into different categorization boundaries.

- **Language Generation** - We consider summarization and data-to-text generation tasks. We use CNN/Dailymail (Hermann et al., 2015), XSUM (Narayan et al., 2018), MultiNews (Fabbri et al., 2019), SAMSum (Gliwa et al., 2019), WebNLG (Castro Ferreira et al., 2020) (English), E2E (Dušek et al., 2019) and CommonGen (Lin et al., 2020) to evaluate our models. For WebNLG, E2E and CommonGen, we use the versions from the GEM benchmark (Gehrmann et al., 2021).

- **Language Generation with Human Evaluation** - We evaluate on a variety of text generation tasks using human evaluation, via the GENIE leaderboard (Khashabi et al., 2021). These tasks include aNLG (Bhagavatula et al., 2019), ARC-DA (Clark et al., 2018), WMT19 (Foundation), and XSUM (Narayan et al., 2018).

- **Language Understanding, Classification and Question Answering** - We use Reading Comprehension, Question Answering, Text Classification and natural language inference datasets. Concretely, we use RACE (Reading comprehension) (Lai et al., 2017), QASC (Khot et al., 2020), OpenBookQA (Mihaylov et al., 2018), TweetQA (Xiong et al., 2019), QuAIL (Rogers et al., 2020), IMDB (Maas et al., 2011), Agnews (Zhang et al., 2015), DocNLI (Yin et al., 2021), Adversarial NLI (Nie et al., 2019), VitaminC (Schuster et al., 2021a), Civil Comments and Wikipedia Toxicity detection datasets (Borkan et al., 2019). We also use standard SuperGLUE (Wang et al., 2019) and GLUE (Wang et al., 2018) datasets.

- **Commonsense Reasoning** - We use HellaSwag (Zellers et al., 2019), SocialIQA/SIQA (Sap et al., 2019), PhysicalIQA/PIQA (Bisk et al., 2020), CosmosQA (Huang et al., 2019), AbductiveNLI (Bhagavatula et al., 2019), CommonsenseQA (Talmor et al., 2018), CommonsenseQA2 (Talmor et al., 2021).

- **Long Range Reasoning** - We use the Scrolls benchmark (Shaham et al., 2022) which comprises of seven component tasks including GovReport (Huang et al., 2021), SumScr (Chen et al., 2021), QMSUm (Zhong et al., 2021), QASPER (Dasigi et al., 2021), NarrativeQA (Kočiský et al., 2018), QuALITY (Pang et al., 2021), and ContractNLI (Koreeda & Manning, 2021).

- **Structured Knowledge Grounding** - We use several component tasks from UnifiedSKG (Xie et al., 2022), namely WikiTQ (Pasupat & Liang, 2015), CompWQ (Talmor & Berant, 2018), FetaQA (Nan et al., 2021), HybridQA (Chen et al., 2020), WikiSQL (Zhong et al., 2017), TabFat (Chen et al., 2019), Feverous (Aly et al., 2021), SQA (Iyyer et al., 2017), MTOP (Li et al., 2020) and DART (Nan et al., 2020). We select datasets that are relatively convenient to perform evaluation and uses mainstream metrics such as accuracy or exact match instead of obscure ones or those that require significant domain specific post-processing.

- **Information Retrieval** - IR is the task of retrieving relevant documents given queries. We use the setup of the latest next generation IR paradigm, i.e., differentiable search index (Tay et al., 2022) for our experiments. We use the same NQ (Kwiatkowski et al., 2019) splits in the DSI paper.

For each dataset, we report the best previous sota result. For generation tasks, we generally report ROUGE-2 following the advice of (Gehrmann et al., 2022). For the rest of the datasets, we report the dominant metric that is reported in prior work. For BLEU scores, we use sacrebleu. For common-sense reasoning tasks, we do not compare against approaches that use external knowledge bases as they are orthogonal and out of scope for this paper. For most part, GLUE is generally considered to be saturated and there are many unpublished results on the GLUE leaderboard. For this reason, we make a very reasonable decision of considering (Raffel et al., 2019) to be the state-of-the-art since we believe that there has not been any real advance on the GLUE benchmark since the T5 model (Raffel et al., 2019). GLUE results, given how saturated it already is, are provided as a reference and should be taken with a pinch of salt.

### B.1.3 SUMMARY OF SUPERVISED FINETUNING RESULTS

This section describes the overview results of our experiments.

Table 11: Summary of UL20B results compared to state-of-the-art. $(l)$ denotes leaderboard submission. $(\sharp)$ denotes the best published we could find on the leaderboard. $(e)$ denotes SOTA used an ensembled approach. Because we evaluate finetuning and in-context trade-offs for SuperGLUE, SuperGLUE scores have their own dedicated section below.

| Dataset | Metric | Eval | Sota Reference | SOTA | Ours |
|---|---|---|---|---|---|
| CNN/DM | Rouge-2 | Test | Zoph et al. | 21.7 | **21.9** |
| XSUM | Rouge-2 | Test | Zoph et al. | **27.1** | 26.6 |
| MultiNews | Rouge-2 | Test | Xiao et al. | 21.1 | **21.7** |
| SAMSum | Rouge-2 | Test | Narayan et al. | 28.3 | **29.6** |
| Gigaword | Rouge 2 | Test | Aghajanyan et al. | **20.7** | **20.7** |
| WebNLG (en) | Rouge-2 | Test | Bakshi et al. | 53.5 | **55.4** |
| E2E-NLG | Rouge-2 | Test | Xue et al. | 45.8 | **46.5** |
| CommonGen | Rouge-2 | Dev | Gehrmann et al. | 32.5 | **37.4** |
| Schema-Guided Dialog | Rouge-2 | Test | Gehrmann et al. | 36.8 | **44.1** |
| | | | | | |
| GENIE - aNLG | Human (H) | Test | Khashabi et al. | 76.0 | **77.0**$^{(l)}$ |
| GENIE - ARC-DA (w/o IR) | Human | Test | Khashabi et al. | **72.0** | **72.0**$^{(l)}$ |
| GENIE - WMT19 | Human | Test | Khashabi et al. | **71.0** | 67.0$^{(l)6}$ |
| GENIE - XSUM | H-Overall | Test | Clive et al. | **51.0** | 50.0$^{(l)}$ |
| GENIE - XSUM | H-Concise | Test | Clive et al. | **53.0** | **53.0**$^{(l)}$ |
| GENIE - XSUM | H-Fluency | Test | Clive et al. | 51.0 | **52.0**$^{(l)}$ |
| GENIE - XSUM | H-No-Hallucination | Test | Clive et al. | 53.0 | **54.0**$^{(l)}$ |
| GENIE - XSUM | H-Informativeness | Test | Clive et al. | **49.0** | **49.0**$^{(l)}$ |
| | | | | | |
| SIQA | Accuracy | Test | Lourie et al. | 83.2 | **83.3**$^{(l)}$ |
| PIQA | Accuracy | Test | Lourie et al. | 90.1 | **90.7**$^{(l)}$ |
| CSQA | Accuracy | Dev | Lourie et al. | 79.1 | **84.9** |
| CSQA2 | Accuracy | Test | Lourie et al. | 69.6$^{(\sharp)}$ | **70.1**$^{(l)}$ |
| QASC (w/o IR) | Accuracy | Dev | Khashabi et al. | 81.8 | **83.8** |
| QASC (w IR) | Accuracy | Test | Khashabi et al. | 89.6 | **90.7**$^{(l)}$ |
| TweetQA | BLEU-1 | Dev | Khashabi et al. | 77.5 | **78.4** |
| QuAIL | Accuracy | Test | Khashabi et al. | 74.2 | **87.2** |
| AdversarialQA (Bert) | F1 | Dev | Khashabi et al. | 53.6 | **70.1** |
| AdversarialQA (Roberta) | F1 | Dev | Khashabi et al. | 45.5 | **57.5** |
| AdversarialQA (Bidaf) | F1 | Dev | Khashabi et al. | 71.5 | **77.5** |
| MCScript | Accuracy | Test | Khashabi et al. | 95.1 | **97.3** |
| MCScript 2.0 | Accuracy | Test | Khashabi et al. | 94.6 | **97.9** |
| RACE | Accuracy | Test | Shoeybi et al. | **90.9**$^{(e)}$ | 90.9 |
| DREAM | Accuracy | Test | Wan | **91.8** | **91.8** |
| OBQA | Accuracy | Test | Khashabi et al. | **87.2** | **87.2**$^{(l)}$ |

Continued on next page

---

[6]This task is German-to-English translation. Our submission is pretrained on only English C4 then finetuned on only the provided WMT19 data (no German pretraining, parallel data or backtranslation.)

**Table 11 – continued from previous page**

| | | | | | |
|---|---|---|---|---|---|
| CosmosQA | Accuracy | Test | Lourie et al. | **91.8** | $91.6^{(l)}$ |
| Winogrande XL | Accuracy | Test | Lourie et al. | **91.3** | $90.1^{(l)}$ |
| | | | | | |
| DocNLI | Accuracy | Test | Qin et al. | 76.9 | **88.2** |
| AdversarialNLI (r3) | Accuracy | Test | Wang et al. | 47.7 | **53.5** |
| VitaminC | Accuracy | Test | Schuster et al. | 90.8 | **91.1** |
| Hellaswag | Accuracy | Test | Lourie et al. | 93.9 | $\mathbf{94.1}^{(l)}$ |
| QQP | F1 | Dev | Raffel et al. | 90.1 | **90.6** |
| QNLI | Accuracy | Dev | Raffel et al. | 96.1 | **96.5** |
| CoLA | Matthews | Dev | Raffel et al. | 68.6 | **71.5** |
| STSB | Spearman | Dev | Raffel et al. | 92.1 | **92.3** |
| AbductiveNLI | Accuracy | Test | He et al. | $\mathbf{89.8}^{(\sharp)}$ | $87.5^{(l)}$ |
| MultiNLI | Accuracy | Dev | Raffel et al. | **92.1** | 91.9 |
| | | | | | |
| IMDB | Accuracy | Test | Yang et al. | 96.2 | **97.3** |
| AgNews | Error | Test | Yang et al. | 4.45 | **4.42** |
| Civil Comments | F1 | Dev | Tay et al. | 87.8 | **87.9** |
| Wikipedia Toxicity | F1 | Dev | Tay et al. | 96.5 | **97.0** |
| SST-2 | Acc | Dev | Raffel et al. | **97.3** | 97.0 |
| | | | | | |
| Scrolls Challenge | Aggregate | Test | Shaham et al. | 29.2 | $\mathbf{37.9}^{(l)}$ |
| SumScr | Rouge (Avg) | Test | Shaham et al. | 16.3 | $\mathbf{20.0}^{(l)}$ |
| QMSum | Rouge (Avg) | Test | Shaham et al. | 19.9 | $\mathbf{20.0}^{(l)}$ |
| QASPER | F1 | Test | Shaham et al. | 26.6 | $\mathbf{37.6}^{(l)}$ |
| NarrativeQA | F1 | Test | Shaham et al. | 18.5 | $\mathbf{24.2}^{(l)}$ |
| QUALITY | EM | Test | Shaham et al. | 26.0 | $\mathbf{45.8}^{(l)}$ |
| ContractNLI | EM | Test | Shaham et al. | 77.4 | $\mathbf{88.7}^{(l)}$ |
| GovRep | Rouge (Avg) | Test | Shaham et al. | **37.2** | $36.2^{(l)}$ |
| | | | | | |
| WikiTQ | Accuracy | Test | Xie et al. | 49.3 | **54.6** |
| CompWebQ | Accuracy | Test | Xie et al. | 73.3 | **75.9** |
| FetaQA | BLEU-4 | Test | Xie et al. | 33.4 | **35.8** |
| HybridQA | Accuracy | Dev | Eisenschlos et al. | 60.8 | **61.0** |
| WikiSQL | Accuracy | Test | Xie et al. | 86.0 | **87.3** |
| TabFat | Accuracy | Test | Xie et al. | 83.4 | **87.1** |
| Feverous | Accuracy | Dev | Xie et al. | 82.4 | **85.6** |
| SQA | Sent.Acc | Test | Xie et al. | 62.4 | **70.5** |
| MTOP | Match | Test | Xie et al. | 86.8 | **87.5** |
| DART | BLEU-4 | Test | Aghajanyan et al. | 47.2 | **50.4** |
| | | | | | |
| DSI-NQ | HITS@10 | Dev | Tay et al. | 70.3 | **73.8** |

### B.1.4 TRADEOFFS BETWEEN FINETUNING AND PROMPT-BASED ZERO-SHOT LEARNING (SUPERGLUE)

In this section, we explore finetuning and in-context learning trade-offs on the SuperGLUE benchmark. We conduct experiments on SuperGLUE with UL20B. While UL20B does not achieve SOTA on this benchmark, we note that UL20B at least remains competitive and outperforms T5-11B. This section reassures that UL2 indeed scales and matches/slightly outperforms T5-11B on SuperGLUE (while strongly outperforming T5-XXL on many other in-context tasks). UL20B still lacks behind the SOTA model ST-MoE-32B given two main reasons. Firstly, ST-MoE-32B has 200B+ parameters and is costs equivalent to a 32B dense model. Secondly, ST-MoE-32B is trained solely on span corruption using an encoder-decoder architecture which is known to be very advantageous on NLU finetuning.

Table 12: Results on SuperGLUE dev set. We compare with T5-11B (Raffel et al., 2019), ST-MoE-32B (Zoph et al., 2022) and PaLM-8B, PaLM-62B and PaLM-540B (Chowdhery et al., 2022). Scores reported are the peak validation scores per task.

| Model | BoolQ | CB | CoPA | MultiRC | Record | RTE | WiC | WSC | Avg |
|---|---|---|---|---|---|---|---|---|---|
| PaLM 62B | 90.6 | 96.4/95.7 | 98.0 | 87.7/61.9 | 93.0/92.4 | 89.5 | 75.9 | 96.2 | 89.2 |
| PaLM 540B | 92.2 | 100/100 | 100 | 90.1/69.2 | 94.0/94.6 | 95.7 | 78.8 | 100 | 92.6 |
| ST-MoE 32B$_{269B}$ | **93.1** | **100/100** | **100** | **90.4/69.9** | **95.0/95.6** | **95.7** | **81.0** | **100** | **93.2** |
| PaLM 8B | 87.6 | 96.4/92.1 | 86.0 | 81.6/64.0 | 89.7/89.3 | 84.5 | 73.4 | 88.5 | 83.4 |
| T5 11B | 90.8 | 94.9/96.4 | 98.0 | 87.4/66.1 | 93.8/93.2 | 93.9 | 77.3 | 96.2 | 89.9 |
| UL2 20B | 90.8 | 98.7/98.2 | 99.0 | 88.4/64.8 | 93.7/93.2 | 92.1 | 77.3 | 98.1 | 90.7 |

Table 13: Results on zero-shot learning on SuperGLUE dataset. We compare with GPT-3, GLaM and PaLM (Chowdhery et al., 2022). We also include models that are relatively compute-matched with UL20B such as T5-XXL with LM adaptation (Lester et al., 2021), GPT-3 13B and GLaM-8B dense. Notably, UL20B outperforms GPT-3 175B and all other models in a similar compute class on average score.

| Model | BoolQ | CB | RTE | ReCORD | WSC | WiC | COPA | MultiRC | Avg |
|---|---|---|---|---|---|---|---|---|---|
| ST-MoE-32B (269B) | 40.8 | 41.1 | 52.7 | 50.0 | 57.5 | 50.0 | 56.0 | 30.3 | 47.6 |
| GPT-3 175B | 60.5 | 46.4 | 63.5 | 90.2 | 65.4 | 0.0 | 91.0 | 72.9 | 61.2 |
| GLaM-MoE 1.2T | 83.0 | 33.9 | 68.8 | 90.3 | 84.9 | 50.5 | 90.0 | 45.1 | 68.3 |
| PaLM 540B | 88.0 | 51.8 | 72.9 | 92.9 | 89.1 | 59.1 | 93.0 | 83.5 | 78.8 |
| T5-XXL | 44.3 | 37.5 | 48.8 | 85.8 | 59.3 | 50.9 | 70.0 | 23.0 | 52.5 |
| GPT-3 13B | 66.2 | 19.6 | 62.8 | 89.0 | 64.4 | 0.0 | 84.0 | 71.4 | 57.2 |
| GLaM-Dense 8B | 73.6 | 33.9 | 44.0 | 89.2 | 80.7 | 44.0 | 86.0 | 39.0 | 61.3 |
| GLaM-MoE 64E | 72.2 | 40.7 | 60.3 | 88.9 | 81.8 | 49.5 | 86.0 | 52.4 | 66.5 |
| PaLM-Dense 8B | 68.3 | 41.1 | 54.2 | 87.8 | 78.9 | 47.0 | 86.0 | 47.5 | 63.9 |
| UL2 20B (*single ckpt*) | 63.1 | 41.1 | 60.7 | 88.1 | 79.9 | 49.8 | 85.0 | 36.2 | 63.0 |
| UL2 20B (*best*) | 63.1 | 50.0 | 60.7 | 88.1 | 80.6 | 55.2 | 88.0 | 36.2 | 65.2 |

### B.1.5 GENERATIVE FEW-SHOT: XSUM SUMMARIZATION

Finally, we conduct additional few-shot in-context one-shot learning using the XSum dataset We compare our model with the baseline T5-XXL, T5-XXL with LM Adaptation (Lester et al., 2021), LaMDA 137B (Thoppilan et al., 2022), and PaLM (8B, 62B, 540B) (Chowdhery et al., 2022). We run T5-XXL ourselves in the same experimental setup but report results from (Chowdhery et al., 2022) for the other models.

Table 14: Results on One-Shot Summarization on XSUM.

| Model | Rouge-1 | Rouge-2 | Rouge-L |
|---|---|---|---|
| LaMDA 137B | - | 5.4 | - |
| PaLM 62B | - | 11.2 | - |
| PaLM 540B | - | **12.2** | - |
| PaLM 8B | - | 7.9 | - |
| T5 XXL 11B | 0.6 | 0.1 | 0.6 |
| T5 XXL 11B + LM | 13.3 | 2.3 | 10.7 |
| UL2 20B | **25.5** | **8.6** | **19.8** |

Table 14 reports results on 1-shot summarization. We note that T5-XXL performs poorly on this task. Even with LM adaptation, the Rouge-2 score is only 2.3, which substantially lacks behind decoder-only causal language models (e.g., PaLM 8B models). Notably, the off-the-shelf T5-XXL is not able to generate meaningful summaries even with prompting just because it is only trained with span corruption so it is intuitive that some form of adaptation is required for generative few-shot settings. Here is is worth that the performance of UL2 20B is about 3x the performance of

LM adapted T5 XXL model. Moreover, UL2 20B outperform LaMDA 137B and has improved performance compared to PaLM 8B. The best result, however, is still the larger 540B and 62B PaLM models.

### B.2 UL2 FOR CHAIN-OF-THOUGHT PROMPTING

It has recently been shown that language models at scale can perform multi-step reasoning tasks such as math word problems or commonsense reasoning via *chain-of-thought prompting*, which prompts the model to generate a step-by-step reasoning path before giving the final answer (Wei et al., 2022b). Notably, chain-of-thought (CoT) prompting does not require any additional fine-tuning of the model.

A crucial consideration of CoT prompting is that it is an emergent ability of scale (Wei et al., 2022a)—it requires a sufficiently large language model to improve performance, and actually hurts performance for small language models. Hence, the successful use cases of chain-of-thought prompting use either LaMDA 137B (Thoppilan et al., 2022), PaLM 540B (Chowdhery et al., 2022), or OpenAI models (Brown et al., 2020; Ouyang et al., 2022). These models, however, are compute intensive and not available as public checkpoints.

Here we demonstrate that UL2 20B can successfully leverage CoT prompting to solve multi-step arithmetic and commonsense tasks. We use the same benchmark tasks and prompts from Wei et al. (2022b). In Table 15 below, we see that on five arithmetic reasoning datasets, CoT prompting outperforms standard prompting (directly outputting the answer without a chain of thought) for UL2 20B. Similar to Wei et al. (2022b), we also show that CoT prompting can be augmented by using an external calculator ("calc.") to perform arithmetic computational only $(+, -, \times, /)$ to further improve performance by a large margin. In addition, we add self-consistency (Wang et al., 2022b) (denoted as "SC") on top of CoT prompting and observed significant gains consistently across all benchmarks, with an average improvement of 22.5% compared to standard prompting.

Table 15: Chain-of-thought prompting and self-consistency (SC) results on five arithmetic reasoning benchmarks. GSM8K: (Cobbe et al., 2021). SVAMP: (Patel et al., 2021). ASDiv: (Miao et al., 2020). AQuA: (Ling et al., 2017). MAWPS: (Koncel-Kedziorski et al., 2016).

| Model | GSM8K | SVAMP | ASDiv | AQuA | MAWPS | Average |
|---|---|---|---|---|---|---|
| UL2 20B: standard prompting | 4.1 | 10.1 | 16.0 | 20.5 | 16.6 | 13.5 |
| UL2 20B: CoT prompting | 4.4 | 12.5 | 16.9 | 23.6 | 19.1 | 15.3 |
| UL2 20B: CoT prompting + calc. | 6.9 | 28.3 | 34.3 | 23.6 | 42.7 | 27.2 |
| UL2 20B: CoT prompting + calc. + SC | **10.2** | **41.4** | **43.5** | **26.9** | **57.9** | **36.0** |

In addition to arithmetic reasoning, Table 16 shows the performance of CoT prompting using UL2 20B compared to standard prompting on five commonsense reasoning benchmarks. CoT prompting plus self-consistency outperforms standard prompting in four of the five benchmarks, with an average improvement of 14.4%.

Table 16: Chain-of-thought prompting and self-consistency (SC) results on five commonsense reasoning benchmarks. CSQA: (Talmor et al., 2019). StrategyQA: (Geva et al., 2021). Date Understanding and Sports Understanding: (Srivastava et al., 2022). ARC-easy/challenge: (Clark et al., 2018).

| Model | CSQA | StrategyQA | Date | Sports | ARC-e | ARC-c | Average |
|---|---|---|---|---|---|---|---|
| UL2 20B: standard prompting | 34.2 | **59.0** | 13.5 | 57.9 | 32.2 | 29.8 | 37.8 |
| UL2 20B: CoT prompting | 51.4 | 53.3 | 14.0 | 65.3 | 61.6 | 42.9 | 48.1 |
| UL2 20B: CoT prompting + SC | **55.7** | 54.9 | **16.3** | **66.8** | **69.8** | **49.5** | **52.2** |

Overall, we have shown that whereas prior CoT work have required large pre-trained models such as PaLM 540B, UL2 20B is a relatively smaller model that can also perform multi-step reasoning. We hypothesize that the mixture of denoisers may contribute to the ability of UL2 to leverage CoT

prompting at 20B parameters, although we leave further investigation of what unlocks emergent chain-of-thought reasoning to future work.

## C    REFERENCE BACKGROUND: PRE-TRAINED LANGUAGE MODELS

In this section, we discuss background surrounding pretrained language models, pretraining objectives and other unified pretraining proposals.

### C.1    PRE-TRAINED LANGUAGE MODELS

Learning pre-trained representations for language is a far-reaching pillar of modern NLP research, dating back to (Mikolov et al., 2013; Pennington et al., 2014; Neumann et al., 2018; Dai & Le, 2015; Howard & Ruder, 2018). The first pre-trained Transformer, GPT, was proposed by (Radford et al., 2019) and was trained as a causal language model. Subsequently, BERT (Devlin et al., 2018) demonstrated the importance of bidirectional modeling for many downstream tasks. BERT introduced masked language modeling (MLM), a denoising objective that reconstructs the input in-place using bidirectional receptive fields. XLNet Yang et al. (2019) introduced the Permutation Language Modeling to account for dependencies between masked tokens during training. A number of additional papers (e.g., RoBERTA (Liu et al., 2019), SpanBERT (Joshi et al., 2020)) suggested further improvements to the pre-training process.

At the same time, two-stack encoder-decoder architectures such as T5 (Raffel et al., 2019) gained popularity due to their improved performance on classification and sequence-to-sequence ("seq2seq") tasks. However, so far, these models have shown limited performance on open-text generation and prompt-based inference (i.e., in-context learning), which motivates the use of decoder-only models that are trained with different objectives (e.g., GPT-3 (Brown et al., 2020), GLaM (Du et al., 2021), LaMDa (Thoppilan et al., 2022) and PaLM (Chowdhery et al., 2022)). In this work, we aim to bridge the performance gap between the two by means of a general training paradigm that suits both architectures.

**Decoder-only vs Encoder-only**    The key similarities of decoder-only and encoder-only architectures is that decoder-only architectures operate with an *input-to-target* paradigm or *targets-only* paradigm if CausalLM is used over PrefixLM used. For both architectures, the objective is always to predict the next token (LM) and are therefore autoregressive models. Notably this is different from position-wise masked LM denoising (sometimes known as *autoencoding*), which have been popularized by encoder-only BERT-style models. These class of models are very restricted in their generative capabilities. On top of that, task specific classification heads are also typically employed for downstream tasks. Because of the cumbersomeness of task specific classification heads, we strongly do not recommend using this class of autoencoding models moving forward and consider them somewhat deprecated. Caveats do apply. For instance, regression is the probably only reason why one would add a task specific head (Lees et al., 2022), or to squeeze out some efficiency gains from eliminating a full vocabulary. Either way, one can always start from a encoder-decoder and chop off the decoder later so there is no good reason to use an encoder-only model. Hence the only real objective consideration here is between decoder-only and encoder-decoder architectures.

**Decoder-only vs Encoder-Decoder**    The line between decoder-only and encoder-decoder models is less clear. PrefixLM models are *almost* encoder-decoder models with shared parameters (but not quite). From an inductive bias point of view, there are multiple differences. Encoder-Decoder models process input and targets independently with a different set of parameters. This is a form of sparsity where different set of parameters are used for different tokens. Encoder-Decoder models also have a cross attention component that connects input tokens to target tokens. Meanwhile, decoder-only models process inputs and targets by concatenating them. Hence, the representations of inputs and targets are concurrently build layer by layer as the input/targets propagate up the network. Conversely, the decoder in Encoder-decoder models generally only looks at the fully processed encoder input. Overall, the inductive bias of PrefixLM decoder-only models and Encoder-Decoder models could be pretty similar modulo the subtle differences stated above. The distinct property is that Encoder-Decoder models are generally approximately 2x parameters of a decoder-only model when compute-matched.

**Sparse Models** On a side note, there have also been also an emerging trend of sparse pretrained models that achieve state-of-the-art performance. Sparse mixture-of-expert models such as the Switch Transformer (Fedus et al., 2021), GLaM (Du et al., 2021) and/or GShard (Lepikhin et al., 2020) have also demonstrated a lot of promise. While orthogonal to the topic of pretraining objectives, sparse models achieve a very different flop-per-parameter ratio compared to dense models - a core recurring motif in the debate surrounding encoder-decoder models vs decoder-only models.

## C.2 Pre-training Objectives for Large Language Models

While recent research demonstrates the potential of large *supervised* multi-task pre-training (Aribandi et al., 2021; Sanh et al., 2021; Wang et al., 2022a), most pre-training objectives rely on the vast availability of *unsupervised* data and use self-training techniques. As mentioned above, different architectures typically leverage different objectives. Decoder-only models are typically trained with causal language model objectives to mimic auto-regressive generation (Radford et al., 2019). Raffel et al. (2019) explored many objectives for encoder-decoder models and found span corruption to be effective. (Wang et al., 2022a) conducts a systematic study of different architectures combined with three different pretraining objectives (causal LM, prefixLM and span corruption) and analyzed their impact on zero-shot generalization. Related to our proposed X-denoisers, (Wettig et al., 2022) studies the effect of corruption rate in BERT-style masked language modeling and hypothesizes that this improves sample efficiency along with benefitting larger models. Notably, the benefits of heightened corruption rates as a *standalone* denoiser is still unclear, as noted by (Raffel et al., 2019) and also apparent in our own ablations. Pre-training (or denoising) is generally applied on the subword level (Raffel et al., 2019; Devlin et al., 2018) but it is worth to note that it has also been applied on the character or byte-level (Xue et al., 2021; Tay et al., 2021c). In these setups, the corrupted spans are generally much larger than subword-based denoising.

## C.3 Unified Pre-training Proposals

UniLM (Dong et al., 2019) proposed to train on multiple language modeling objectives using a single Transformer model. Specifically, UniLM trains on unidirectional LM, bidirectional LM and seq2seq LM. This is quite similar to combining auto-regressive LMs with BERT and prefix-LM models. Notably, UniLM trains using a cloze-type formulation which adds explicit mask tokens to the inputs. Losses are then computed by the difference of the predicted token and target token in a position-wise fashion. Aside from pretraining unification, there have been a recent trend of thematic unification, i.e., unifying common tasks into one model. Examples of these include UNICORN (Lourie et al., 2021) for commonsense reasoning, UnifiedQA (Khashabi et al., 2020; 2022) for question answering, Programming Puzzles (Schuster et al., 2021b) for problem solving, and UnifiedSKG (Xie et al., 2022) for Structured Knowledge Grounding.

## C.4 Implementation Details and UL2 code

This section aims to give more insight to how UL2 pretraining is implemented. Our implementation is actually pretty simple. It is simply a mixture of different pretraining objectives that is implemented in seqio[7]. Most of our experiments were run with simply mixing different seqio tasks with seqio's Mixture Registry. However, one could also implement a generalized UL2 objective with the following function which could be cleaner.

```
def ul2_objective(dataset: tf.data.Dataset,
                  sequence_length: seqio.preprocessors.SequenceLengthType,
                  output_features: seqio.preprocessors.OutputFeaturesType,
                  use_prefix_lm_task: bool = False,
                  rates: Optional[Sequence[float]] = None,
                  mean_noise_span_lengths: Sequence[float] = (3.0,),
                  noise_densities: Sequence[float] = (0.15,),
                  shard_ds: bool = True,
                  optional_task_prefixes: Optional[Sequence[str]] = None,
                  input_feature_key: str = "inputs",
                  merge_examples_to_reduce_padding: bool = True,
```

---

[7] https://github.com/google/seqio

```python
                reserved_for_packing: bool = None,
                seed: int = 7) -> tf.data.Dataset:
  """UL2-like pre-training objectives.

  This preprocessor amounts to calling the `span_corruption` function
      several
  times with different values of `noise_density` and
      `mean_noise_span_length`.
  We either shard or copy the dataset, then apply each function to each
      shard.
  Add S-denoising (prefixLM) using use_prefix_lm_task.

  Args:
    dataset: A tf.data.Dataset with dictionaries containing the key
      `input_feature_key`.
    sequence_length: dict mapping of feature key to int length for that
        feature.
    output_features: mapping of keys to features.
    use_prefix_lm_task: <bool> If True, include PrefixLM in the task mix.
    rates: <Optional<List<float>> List of rates per task. If None, tasks
        are
          sampled uniformly.
    mean_noise_span_lengths: List of mean number of tokens per masked
        span per
      example.
    noise_densities: List of what fraction of the tokens to mask.
    shard_ds: <bool> If True, shard dataset per objective.
    optional_task_prefixes: <Optional<list<str>> Strings to prepend for
        each
                        corruption scheme. NOTE: If including prefixLM task,
                        it must be the last prefix.
    input_feature_key: which feature to use from the dataset as the input
        text
      tokens.
    merge_examples_to_reduce_padding: if True, combines multiple input
        examples
      to reduce padding.
    reserved_for_packing: if specified, reduces the desired inputs length
        by the
      specified amount to enable multiple examples to be packed together
      downstream.
    seed: tf.int64 for controlling the random choice of spans.

  Returns:
    a dataset
  """

  if optional_task_prefixes: # Ensure each task has a prefix.
    num_tasks = len(noise_densities) + int(use_prefix_lm_task)
    valid_number_of_prefixes = num_tasks == len(optional_task_prefixes)
    if not valid_number_of_prefixes:
      raise ValueError("Number of task prefixes must match number of
          tasks.")
  inputs_length = sequence_length[input_feature_key]
  input_lengths, targets_lengths = [], []
  sequence_lengths = {x: y for x, y in sequence_length.items()}
  if reserved_for_packing:
    inputs_length -= reserved_for_packing
    for x, y in sequence_length.items():
      sequence_lengths[x] = y - reserved_for_packing
  hyperparams = list(zip(mean_noise_span_lengths, noise_densities))
  for mean_noise_span_length, noise_density in hyperparams:
    input_length, targets_length =
        t5.data.preprocessors.random_spans_helper(
        extra_tokens_per_span_inputs=1,
```

```
      extra_tokens_per_span_targets=1,
      inputs_length=inputs_length,
      mean_noise_span_length=mean_noise_span_length,
      noise_density=noise_density)
  input_lengths.append(input_length)
  targets_lengths.append(targets_length)

  if sequence_length["targets"] < targets_length:
    upper_bound = max(targets_lengths)
    raise ValueError(
        f'Expected max targets length for span corruption ({upper_bound})
             is '
        f'greater than configured targets length '
        f"({sequence_length['targets']})")

ds = dataset
ds = t5.data.preprocessors.select_random_chunk(
    ds,
    output_features=output_features,
    feature_key="targets",
    max_length=65536)
if merge_examples_to_reduce_padding:
  ds = t5.data.preprocessors.reduce_concat_tokens(
      ds, feature_key="targets", batch_size=128)
num_shards = len(input_lengths) + int(use_prefix_lm_task)
if shard_ds:
  ds_shards = [ds.shard(num_shards, i) for i in range(num_shards)]
else:
  ds_shards = [ds for _ in range(num_shards)]
processed_ds = []
hyperparams = zip(input_lengths, hyperparams, range(num_shards))
for input_length, (noise_span_length, noise_density), i in hyperparams:
  ds = ds_shards[i]
  ds = t5.data.preprocessors.split_tokens(
      ds,
      feature_key="targets",
      min_tokens_per_segment=None,
      max_tokens_per_segment=input_length)
  ds = t5.data.preprocessors.denoise(
      ds,
      output_features,
      inputs_fn=t5.data.preprocessors.noise_span_to_unique_sentinel,
      targets_fn=t5.data.preprocessors.nonnoise_span_to_unique_sentinel,
      noise_density=noise_density,
      noise_mask_fn=functools.partial(
          t5.data.preprocessors.random_spans_noise_mask,
          mean_noise_span_length=noise_span_length),
      input_feature_key=input_feature_key)
  if optional_task_prefixes:
    ds = prepend_prompt(
        ds,
        output_features,
        prompt_mode=optional_task_prefixes[i],
        mode=optional_task_prefixes[i])
  processed_ds.append(ds)
if use_prefix_lm_task:
  ds = ds_shards[-1]
  ds = t5.data.preprocessors.prefix_lm(ds, sequence_lengths,
      output_features)
  if optional_task_prefixes:
    ds = prepend_prompt(
        ds,
        output_features,
        prompt_mode=optional_task_prefixes[-1],
        mode=optional_task_prefixes[-1])
```

```
  processed_ds.append(ds)

ds = tf.data.experimental.sample_from_datasets(processed_ds, rates,
    seed)
return ds
```

## C.5 DETAILS OF SUPERVISED FINETUNING SOTA RUNS

Most of our supervised finetuning runs were finetuned as single tasks. The only exception was that:

- We finetuned GLUE as a single mixture with proportionate sampling. This has become standard and defacto setup (Raffel et al., 2019; He et al., 2022; Tay et al., 2020; 2021b).
- We finetuned SuperGLUE as a single mixture which is also a standard setup these days (Fedus et al., 2021; Raffel et al., 2019; Chowdhery et al., 2022).
- SIQA, PIQA, AbductiveNLI, Winogrande XL and CosmosQA were co-trained in a proportionate mixture similar to (Lourie et al., 2021) under the Rainbow benchmark.
- For CSQA, CSQA2. OBQA, and ARC-DA we co-trained with the rainbow mixture to obtain results on these three datasets.
- All other tasks were single-task finetuned.

## C.6 QUALITATIVE SAMPLES

This section lists some qualitative samples that are randomly sampled (and not cherrypicked) on 1shot XSUM summarization.

---

**Summarize:** John Martin, 48, denies murdering Russian pianist Natalia Strelchenko, 38, at their Manchester home on their second wedding anniversary last August. ; Manchester Crown Court heard he took the diazepam believing it to be his prescribed anti-depressant medication. ; He said "everything is very blurry". ; It is alleged the double bass player strangled and beat Ms Strelchenko, who was also known by the surname Strelle, in a loss of temper at their home in Newton Heath on 30 August. ; Giving evidence, Mr Martin, also known as Jon Skogsbakken, said he had been suffering with depression and that he had unwittingly taken unmarked diazepam tablets for around six weeks. ; He said he "guessed" they had been prescribed to his wife. ; Mr Martin told the jury he had suffered with depression since 2005 and his GP would prescribe him Escitalopram. ; He told the court that taking the diazepam was "the biggest mistake of my life". ; Mr Martin said: "I would never have taken the two together, absolutely not", if he was aware the tablets could have a "paradoxical and aggressive response" and when taken with alcohol it could increase the effects. ; The court heard following an argument with Ms Strelchenko, he drank four cans of cider at their home before going to the working men's club where he had another pint of cider and then returned home. ; He said: "I just sat down on the floor and poured myself a glass of wine. I started to drink it. Natalia was angry because I was drinking wine and she tried to take it away from me. ; "Everything is very blurry for me. I can't recall anything after that point. ; "I still love her very much. I would never wish to do such terrible things to her. I recall virtually nothing." ; Mr Denney QC said: "Do you accept that on evidence you killed Natalia?" ; Mr Martin replied: "Well according to what the witnesses described...I perfectly understand that I must be the man behind it." ; Mr Martin, who had worked for the computer firm IBM in his home country of Norway, met Ms Strelchenko in 2007. ; She had performed piano recitals at New York's Carnegie Hall and London's Wigmore Hall. ; The trial continues.
**Output:** A man has denied murdering his Russian pianist wife after taking a cocktail of drugs and alcohol.

---

**Summarize:** The pair, dressed in Team GB tracksuits and wearing plastic medals, admitted to Sky News they "blagged" their way on to the float and joined the celebrations. ; The float was carrying Britain's gold medal-winning women's hockey team and athletes from the modern pentathlon. ; The BOA said: "It didn't spoil anyone's enjoyment of a wonderful event." ; "We are aware of the matter and are disappointed anyone would want to detract from the athletes' celebration," it added. ; Four hundred athletes took part in the Manchester celebrations with more 150,000 people lining the streets. ; A second parade is to take place in Trafalgar Square in London from 13:00 BST after which the athletes will visit Buckingham Palace. ; Greater Manchester Police said it did not provide security for the floats and had received no reports regarding the incident. ; Manchester City Council is yet to comment. ; Team GB set an Olympic record in Rio 2016 by winning 67 medals - two more than they managed at London 2012. ; The Paralympians also beat their previous performance with 147 medals - 27 more than four years ago. **Output:** Two men have been arrested after attempting to join the Olympic and Paralympic parade in Manchester.

---

**Summarize:** Laura Stewart, 20, was walking on North Hanover Street with her friend Mhairi Convy, 18, when they were hit by a Range Rover on 17 December. ; The funeral of Ms Stewart, an accounts student from Cumbernauld, was held at Our Lady and St Helen's Parish Church in her home town. ; About 200 people attended the ceremony to pay their respects. ; Outside the church floral tributes were laid, including the name "Laura" written out in red and white flowers opposite the word "sister" in purple flowers. ; Tributes left with the flowers included a card saying "with all our love for always, mum and dad" and another to a "beautiful" girl who was a "loving friend and sister". ; The order of service showed a photo of the student with the words "Courage grows strong at a wound" - the motto of the Stewart clan. ; Mourners wept as the white coffin with purple handles was carried out of the church to the hymn Walk With Me Oh My Lord. ; Father Gerald Sharkey led the procession out of the church through the snow. The body was then driven to Eastfield Cemetery in Cumbernauld for a private burial. ; Ms Stewart and Ms Convy, of Lennoxtown,

studied at Glasgow College of Commerce. ; Following the crash, the two women were taken to Glasgow Royal Infirmary, where they died. ; A 36-year-old male pedestrian and the 50-year-old male driver of the Range Rover were also treated in hospital after the crash. ; Strathclyde Police are investigating the deaths.
**Output:** The funeral of a student who died in a car crash in Glasgow has taken place.

---

**Summarize:** Eleven of Scotland's 14 territorial health boards were hit by the "ransomware" attack linked to other IT attacks around the world. ; The hack has encrypted information on NHS computers, denying access unless a payment is made. ; First Minister Nicola Sturgeon said work to fix the systems affected had entered a "recovery phase" ; She called the attack "hugely concerning" but said there was no evidence as yet to suggest outdated computer systems were responsible for the breach. ; Scotland's Health Secretary Shona Robison said there was a "level of confidence" that systems would be back up and running by Monday and that no breach of patient confidentiality had been detected. ; The health boards which have been affected are: ; The Scottish Ambulance Service has also been affected, along with NHS National Services Scotland. ; The incidents are thought to be part of a wider attack affecting organisations in about 100 countries around the world. ; IT problems have also caused disruption in about 30 health authorities in England, while the NHS in Wales and Northern Ireland are so far unaffected. ; The Scottish government said most incidents had been confined to desktop computers in GP surgeries, dental practices and other primary care centres. ; A spokesman said the only acute hospital sites so far affected had been in NHS Lanarkshire. ; BBC Scotland understands that computer systems at Hairmyres Hospital in East Kilbride were compromised. ; Ms Robison told the BBC's Good Morning Scotland programme that IT specialists had been working non-stop to get GP systems back up and running. ; She said: "People are working very, very hard and have worked through the night. The update I've got this morning is that we're very much into recovery phase now, with a lot of work going on to get systems back up running. ; "The GP systems, which of course were the main problem across our health boards - work is going on, and there is a level of confidence that many will be back up and running before GP surgeries open on Monday morning." ; Ms Sturgeon told the BBC: "Obviously cyber-attacks of this nature are hugely concerning and I think they underline the vulnerability not just of the public sector, but also of society generally to cyber-attacks, but they also underline the importance of all organisations making sure that they have all appropriate measures in place to protect against those kinds of attack." ; She added: "Obviously there is a lot of investigation into exactly why the health service has been affected in the way it has. I think it's important to stress that this has been a global international attack." ; Shona Robison said NHS Lanarkshire had been more affected in terms of its acute hospitals, but said manual systems had worked safely and insisted that patients had not been negatively impacted by the breach. ; "The intention today is to begin testing those IT systems and to gradually and safely try to bring those back on over the course of the weekend." ; Ms Robison added: "Throughout all of this there's been no breach to patient confidentiality that has been detected to date so patients should be reassured by that." ; Patient services, including emergency services, are continuing to operate across Scotland, the Scottish government confirmed. ; Ms Robison emphasised there had been no impact on the majority of the out-of-hours systems across Scotland, with NHS 24 working as normal along with the Scottish Ambulance Service, where the only issue had been with desktop PCs that were "non-patient facing". ; "All the other parts of the system that people would use over the weekend are working as normal," Ms Robison said. ; In NHS Lanarkshire, non-emergency patients have been urged to stay away from its hospitals as it deals with the ransomware attack. ; Dr Helen Mackie, chief of medical services at Hairmyres Hospital, said staff had reverted to paper and manual records for patients. ; She has urged any patients turning up at the authorities emergency departments to bring their medication with them because medics may have problems accessing their records. ; She said: "Help us by bringing as much information with you, So bring your medicines, bring any information you have about health care. Relatives can really help us as well because they're a wealth of information for us." ; Dr Mackie said the IT issues were leading to the cancellation of planned out-patient appoints for tests such as CT scans. However, she said that any emergency diagnostic tests would continue to take place. ; She added: "It's business as usual in terms of emergency care in our Lanarkshire hospitals. We are asking the public to help us by only coming to emergency departments if it is an emergency. But please be reassured that all our emergency access to treatment and care is up and fully running."
**Output:** Eleven of Scotland's 14 territorial health boards were hit by the "ransomware" attack

linked to other IT attacks around the world.

---

**Summarize:** Two-time major winner McIlroy, 25, and Danish former world number one Wozniacki, 23, announced their engagement on New Year's Eve. ; Media playback is not supported on this device ; "The problem is mine," McIlroy said. "The wedding invitations issued at the weekend made me realise that I wasn't ready for all that marriage entails." ; The couple had been together for more than two years. ; McIlroy is currently at Wentworth for the BMW PGA Championship, the European Tour's flagship event which starts on Thursday. ; Wozniacki is set to compete in the French Open in Paris starting on 25 May. ; In a statement issued through his management company, McIlroy added: "There is no right way to end a relationship that has been so important to two people. ; "I wish Caroline all the happiness she deserves and thank her for the great times we have had." ; "The news has shocked the worlds of both golf and tennis, particularly with the timing of the split. McIlroy is due to start a crucial run of tournaments, while Wozniacki is playing the upcoming French Open." ; Speaking at Wentworth on Wednesday, McIlroy said the decision to end the engagement had been mutual and amicable. ; He said it was a difficult time, but that the move was best for both of them. ; McIlroy won the US Open in 2011 and US PGA Championship the following year. ; Wozniacki, who has reached the final of one grand slam tournament - the US Open in 2009 - is currently ranked 13th in the world. ; When McIlroy announced their engagement, he tweeted: "I have a feeling it is going to be a great year."
**Output:** Rory McIlroy and Caroline Wozniacki have ended their engagement.

---

