# OpenReview forum: "UL2: Unifying Language Learning Paradigms"
_ICLR.cc/2023/Conference — ICLR 2023 poster_

### Official Review · Reviewer_Laoj · 2022-10-23

**Confidence:** 4
**Correctness:** 4
**Technical Novelty And Significance:** 3
**Empirical Novelty And Significance:** 3
**Recommendation:** 8

**Clarity, Quality, Novelty And Reproducibility:**

This paper is well-written and easy to follow. The proposed method is straightforward and effective. The experimental comparisons and analysis are detailed. The pretrained checkpoints will be released for reproducibility.

**Strength And Weaknesses:**

Strengths
1. The idea of this paper is simple, but very effective. Through the MoD method and mode switch, several pre-training tasks with different forms are successfully unified.
2. In our intuition, the various pre-training tasks seem to be independent of each other, and it seems reasonable to pre-train models for these tasks separately. However, the author can jump out of this mindset, observe the common points of these tasks, and propose a unified pre-training model framework by introducing effective methods, and finally achieve good results. Solving big problems with small methods inspires us a lot.
3. This paper is well-written. From the proposal, to the solution, to the experimental setup and results, and finally to scaling to 20B parameters and testing the effectiveness of the chain of thought method.
4. The experiment of this paper is very detailed. Although the main text is only 9 pages, the experimental part of the appendix is as many as 13 pages. It can be seen that the author is very comprehensive when considering the problem and is very patient when researching the problem. Although there are many experimental results, the layout is very neat, well organized, and the levels are rigorous. This scientific research spirit is worth learning.

Weakness:
1. I personally suggest that the authors need to modify Figure 2. The text is too small to recognize each denoiser’s feature.
2. Will the settings of denoisers, like different combinations, different hyper-parameters,  exert huge influence on UL2’s performance? More experiments are required to prove it.
3. In the comparisons between decoder and encoder-decoder architecture, UL2 decoder doesn’t outperform UL2 encoder-decoder. But in the paper, “However, this reinforces our point that the self-supervision objective may be intrinsically more important than the backbone architecture and negotiating architectural choices is mainly about efficiency trade-offs that can be studied independently.” is somewhat confusing. The authors need to give more detailed explanations.


**Summary Of The Paper:**

The authors propose a unified pre-training model framework for the problems of existing pre-training models, which are usually targeted at specific categories of problems and have poor performance in cross-task domains. First, the authors use R-Denoiser, S-Denoiser and X-Denoiser to define multiple pre-training tasks uniformly, and propose Mixture-of-Denoisers (MoD) that frame multiple pretraining tasks as span corruption, diversifies and then mixes them. Then, the authors introduce the notion of paradigm-shifting via mode switching. During pre-training, they feed the model an extra paradigm token, {[R], [S], [X]} that helps the model switch gears and operate on a mode that is more suitable for the given task. Next, the authors do a large number of ablation experiments, and discuss 'Decoder Vs Encoder-Decoder' and 'Is GPT and/or T5 the optimal setup?' questions separately. They experimentally demonstrate the performance of the UL2 model. Finally, the authors also train a 20B parameters UL2 model, and conduct experiments over 50 NLP tasks under the supervised finetuning paradigm. They also apply chain of thought on the model, and show that CoT is also suitable for UL2.

**Summary Of The Review:**

To sum up, this paper is well-written and easy to follow. The proposed method is straightforward and effective. The experimental comparisons and analysis are detailed. I recommend accepting this paper.

---

> ### Author Response · Authors · 2022-11-18
> **Rebuttal**
>
> Thanks for the review!
>
> About your comments:
> 1. About the diagram, we're going to make it bigger for sure but we're working with our in-house designer to modify the diagram. We're trying to make it happen by the revision deadline. But rest assured we will definitely make it bigger in the actual version later! We don't want folks to have to squint so thanks for pointing it out.
> 2. About the extensive hparam search, we have provided some results in the Appendix (Table 9) which we think its quite comprehensive. The current finding is that overall config does change performance quite a bit.
> 3. The discussion about encoder-decoder and decoder-only is discussed in Section C.1 "Decoder-only vs Encoder-Decoder". the main point here is that Encoder-decoders always have twice the number of parameters as decoder-only model at "compute-match" (this is not so intuitive, we know, so bear with us.). The tricky part is whether to match EncDec and Dec at the same params or same compute. For many practitioners, speed and training time is more important (as it also correlates to training cost), therefore we match EncDec and Dec at the same compute. This results in EncDec models having 2x parameters which makes it kind of a "sparse model". Therefore, at certain compute regions, this can lead to EncDec having an inherent advantage which is also why the EncDecs in the main experiments outperform the Dec models. Due to the lack of space, we unfortunately had to move this to the appendix.
>
> Overall, we are happy you enjoyed and appreciated the paper. Thanks for taking the time to review the paper!

---

### Official Review · Reviewer_m24X · 2022-10-26

**Confidence:** 4
**Correctness:** 2
**Technical Novelty And Significance:** 2
**Empirical Novelty And Significance:** 3
**Recommendation:** 3

**Clarity, Quality, Novelty And Reproducibility:**

It is not clearly written how different denoising scheme is combined during training, from the implementation in the appendix it seems that they are equally weighted, however, it is not clearly indicated in the main text. Also the exact model configuration for the smaller models are not specified, which would be hard for reproducing the results.

**Strength And Weaknesses:**

The authors compare UL2 against other objectives at a small scale (~335M parameters) and showed overall better performance, as well as scaling up to 20B parameters and demonstrated better zero-shot performance compared to notable large models like GPT-3.

The paper has a value in terms of the comprehensive comparison across objectives with relevant discussions, however, there are several concerns listed in the following:

1) Evaluation
The evaluation seems a bit arbitrary, also as the author written in appendix, "We arbitrarily select XSUM, ToTTo and SGD from the GEM benchmark". Is there a reason for this "arbitrary" selection instead of testing on more datasets from GEM? Also since the Rouge-L scores are quite low in all the experiments, it would be great to show some qualitative examples to demonstrate that the model generated results are actually meaningful. Because those metrics are not very representative when the score is very low.

The normalized percentage gain is not representative of the overall performance. Look at Table 2 and 3, the major relative gain all comes from the one-shot experiments, in particular, the ones with very bad baseline performance and achieve well above 100% relative performance improvements. However, if we instead look at the actual performance numbers on those tasks, it is also not in a reasonable range. Therefore, I think the gain over pervious models are over claimed. Additionally, the one-shot experiments would exhibit high variance based on the one-shot example selection, it would be great if the authors run with several different samples and report the overall result with the variance.

In addition, there are a couple of notably inconsistent numbers listed below.
- On one shot XSum, SC (i.e., T5, 335M) in Table 2 achieves 7.49 ROUGE-L, while a much larger T5-XXL 11B achieves 0.6 ROUGE-L in Table 6.
- PaLM 8B in Table 6 reports 4.5 ROUGE-2, while the PaLM paper (Table 16) reports 7.9 ROUGE-2. In fact “4.5" appearing in PaLM paper is for WebNLG (ru) task, not XSum.
- Abstract says “tripling the performance of T5-XXL” but UL2 has double the number of parameters. The comparison isn’t entirely fair.

(2) Utility of paradigm tokens are not demonstrated
Authors suggest that each denoising paradigm has a suitable kind of downstream tasks. In particular, it is said that R-denoiser is like T5 objective, and S or X-denoisers are more suitable for generating fluent texts. However, XSUM results in Table 8 suggest otherwise — T5-like R-denoiser is the most suitable. In addition, compare the results with different paradigm tokens and the ones without in table 8 indicates that the paradigm token has minimal (or negative) effect on these tasks, so it is unclear whether those tokens are actually useful.

(3) Technical Novelty
As I understand, UL2 combines different type of span noise with different level of noise and restrictions (e.g. random locations vs. only in the latter part of the sequence). The only thing that differentiate a bit from existing work is the paradigm token, however, the effectiveness of this token is also not clear.




**Summary Of The Paper:**

This paper proposes the mixture-of-denoisers as the training objective for pertained language models. The mixture consists of two types of denoising pradigms of different noise configurations and a prefix language model-like next token prediction. Authors hint that each paradigm is suitable for a particular kind of downstream tasks, such as language understanding and generation. During training, special tokens are prepended to the input such that the paradigm can be switched during test time depending on the downstream tasks.
The authors compare UL2 against other objectives at a small scale (~335M parameters) and showed overall better performance, as well as scaling up to 20B parameters and demonstrated better zero-shot performance compared to notable large models like GPT-3.

**Summary Of The Review:**

Please refer to Strength And Weaknesses

---

> ### Author Response · Authors · 2022-11-18
> **Rebuttal**
>
> Thanks for the reviews and feedback.
>
> Generally, regarding eval, the GEM benchmark is comprehensive and there are so many benchmarks in GEM. Some form of benchmark selection is necessary in this case. The terminology and intention of using “arbitrary” here is just to reflect that we did not run more than the selected tasks and cherry pick the results (which would be bad imo). In fact, those 3 were the only ones we ran for in-context generation. We picked XSUM due to it being just more well-known in general (and we think its a sound benchmark), and then ToTTO and SGD randomly without any prior.
>
> About the XSUM and the PaLM 8b result, thanks for spotting the typo. We have fixed that in the revised version. It is worth noting that PaLM 8B and UL2 20B actually uses the exact same scaling configuration and is approximately compute-matched (aligning with our encoder-decoder vs decoder discussions) that enc-decs are 2x params at compute-matched. We have updated the draft with the new number and the prose accordingly.
>
> About T5 XXL vs UL2 and the conflation of model scale, we have an exact parameter-match and FLOP comparison in Table 10, the result of UL2 is 2.4 vs 0.8 which is also 3x.
>
> About the qualitative evals, we have updated the draft with some qualitative samples of 1-shot XSUM. It is also good to note that we ran our evals on the GENIE benchmark which included *human evals*. So not all evaluation is automatic. This should ease concerns about the qualitative samples and we agree it is a good thing to add.
>
> Overall, in good faith, we fixed some issues you have mentioned that we think was pretty good feedback. I’m not sure what to make about the comment about technical novelty. Instead of trying to argue about some vague definition of “novelty”, I think it might be best (as good scientists) to have a nose for impactful and clear contributions. Wdyt? Sorry for the unsolicited advice but i think this is a pretty good paper that should be accepted.
>
> PS: i didn't mean to write a meta-review for my own paper. :)
>
> Again thanks for taking time to review the paper.

---

> > ### Comment · Reviewer_m24X · 2022-11-20
> > **Thanks for additional qualitative results**
> >
> > Thanks the authors for providing additional qualitative results for 1-shot in-context on XSUM. It is not clear to me from which model are the results obtained? Is it from the 167M decoder only, 335M encoder-decoder or the 20B? Is it possible to have some samples from different sized models?
> >
> > The difference between T5 XXL vs. SC 335M is still unclear, where T5 XXL in-context 1-shot on XSUM is 0.6 and SC (which is T5, right?) is 7.49 on Rouge-L. Why is there such a big difference? Also the smaller models are performing so much better.
> >
> > The usefulness of proposed paradigm token is unclear, on the ablated datasets, it does not seem to make much difference as compared to not using it.
> >
> > Overall, I think there is some value in mix different denoising objectives, however, the effect of the paradigm tokens (i.e. [S], [R], [X]) is unclear. I think the authors needs to more carefully look at the contributions from those components and claim accordingly.

---

### Official Review · Reviewer_9xAc · 2022-11-01

**Confidence:** 4
**Correctness:** 4
**Technical Novelty And Significance:** 3
**Empirical Novelty And Significance:** 3
**Recommendation:** 8

**Clarity, Quality, Novelty And Reproducibility:**

This work is basically incremental (combining existing pretraining objectives), but the paper does a good job at answering research questions regarding the effectiveness of pretraining objectives and model architectures, so I don't see this as a big problem.
The writing is very clear.

**Strength And Weaknesses:**

Strengths:
- The paper's idea is very clear and will be useful for other people in the field.
- The authors will release new pretrained models with strong performance.
- I was initially worried that the authors wouldn't back up all their initial claims, but the experiments and ablation studies support the main arguments extremely well.

Weaknesses:
- The novelty (in the sense of *creativity*) of this paper is limited (but the insights are still useful!).
- There are a couple of typos, but nothing major.
- The ablation study regarding mode switching should be in the main part of the paper (as it's needed to back up one of the main contributions).

**Summary Of The Paper:**

The paper investigates the effect of the concrete pretraining task (and, to a lesser extend), the model architecture, on the effectiveness of the resulting pretrained language representation model. The latter is measured in multiple ways, performing zero-shot, one-shot and normal finetuning experiments.

The paper further introduces Mixture-of-Denoisers (MoD), a pretraining objective that combines multiple pretraining objectives as well as *mode switching*, which associates downstream finetuning with specific ways of pretraining by introducing special tokens, indicating the structure of the (pretraining or finetuning) task to the model.

Comparing T5-style pretraining with GPT-3-style pretraining, the authors find that the former performs best. Their newly proposed model, UL2, however, outperforms the T5-style model in nearly all settings. Looking at the ablation studies, MoD seems clearly beneficial, while mode switching seems helpful (only) in some settings.



**Summary Of The Review:**

The paper asks clear research questions and answers them in a convincing way, so I believe that it should be accepted to the conference.

---

> ### Author Response · Authors · 2022-11-18
> **Rebuttal**
>
> Thanks for the comments and taking time to read/review our paper.
>
> We're glad you liked the paper and enjoyed the insights!

---

### Official Review · Reviewer_qkXg · 2022-11-03

**Confidence:** 4
**Correctness:** 3
**Technical Novelty And Significance:** 3
**Empirical Novelty And Significance:** 3
**Recommendation:** 6

**Clarity, Quality, Novelty And Reproducibility:**

The paper is well written, the experiments is extensive, and it improves over existing multi-task pretraining objectives. Code and model are publicly available.


**Strength And Weaknesses:**

Strength
This paper is well written, the experiments are extensive, and detailed ablation study is presented.

Weakness
While the experiments are extensive, there are questions about the presentation and the conclusion.
- relative performance comparison is not a good metric when the baseline is bad. It leads to 1k+ gain in table 4, skewing the averaging comparison.
- “The best decoder baseline model here is the Prefix-LM decoder model, which is about 10% worse than the T5 baseline.” While this statement is true on average, the performance difference is not true for all tasks, notably LM. It contradicts the universal claim (“ It is clear from these results that encoder-decoder models should be preferred over decoder-only models if and only if there is no concern about storage”) this paper is making.
- “RegardingPrefix-LM pre-training,  it is interesting that Prefix-LM actually outperforms the T5 span corrupt setup by+16.7%.” Is this finding contrary to Raffel et al., 2019?
- Why are some numbers missing in table 6?


**Summary Of The Paper:**

This paper proposes a multi-task pretraining objective that outperforms and unifies prior objectives, it attempts to decouple the modeling architecture and the pretraining objective. The proposed objective shows strong performance gain over SOTA.


**Summary Of The Review:**

While the experiment is extensive, some of the strong claims are not supported by the experiment.

---

> ### Author Response · Authors · 2022-11-18
> **Rebuttal**
>
> Thanks for taking time to review the paper and for the comments and feedback.
>
> Regarding the point about the relative performance comparison, this is normalized in the end so it does not skew any average scores. Otherwise +1k% is going to create a horrible outcome in the average. This is normalized so rest assured.
>
> Nevertheless, the actual raw numbers are still reported so readers can make their own interpretations depending on their use cases.
>
> Comparing many models across different tasks and in some sense, constructing a utility function to describe this comprehensively is difficult. In this case, we used the normalized relative gain (and average) to make comparisons between models for our eventual claim. This is a very hard problem to solve and there are always trade-offs between solutions and some form of interpretation of results is always required. See “the benchmark lottery” which discusses some of these issues too.
>
> The PLM outperforming span corruption overall again applies to a well rounded evaluation based on the tasks we have selected. I believe the original T5 paper mainly considered SuperGLUE, CNN and WMT without considering open ended generation and 1-shot tasks. The inclusion of few-shot tasks likely changed this finding.
>
> Again, the design of the evaluation space is extremely challenging and it’s always up to interpretation and trade-offs.
>
> Numbers in Table 6 are missing because the source papers do not report them. Since Rouge-2 is our main metric (also suggested by the GEM benchmark authors), we think its just okay to have Rouge-2 as the main comparison point.
>
> Thanks for taking the time to review the paper!

---

### Public Comment · ~Stella_Rose_Biderman1 · 2022-11-07
**"Successfully leverag[ing] CoT" claim seems dubious**

You write

> Here we demonstrate that UL2 20B is the first publicly available pre-trained model (without any fine-tuning) to successfully leverage CoT prompting to solve multi-step arithmetic and common-sense tasks.

I have two issues with this claim. Firstly, the performance improvement from CoT is quite small as shown in Table 15. The average improvement is from 13.5 to 15.3, with some improvements being as small as from 4.1 to 4.4. When someone says that they successfully used a technique to solve arithmetic tasks, I expect that to mean that they performed well at the tasks (which UL2 doesn't) and performed much better than they would have without the tech (which is at best non-obvious). Secondly, you repeatedly state variants on "UL2 is the first open source and non-large (<100B) parameter scale model that benefits from CoT reasoning." However I see no evidence for a claim like this in your paper. It is the first open source < 100B parameter model that has been *evaluated using CoT prompting* as far as I am aware, but your claim specifically states that models like EleutherAI's GPT-NeoX-20B, BigScience's T0++, and Meta's OPT-66B don't benefit from CoT prompting. In order to claim this, you need to have evaluated those models and found that they don't benefit from CoT prompting.

---

> ### Author Response · Authors · 2022-11-18
> **Let's think step by step**
>
> Thank you for the comment for reading our paper.
>
> Regarding the average improvement, indeed using basic CoT only improves by a few percent, but when combined with an external calculator (as done in Cobbe et al., 2021) the improvement is 13.7% and jumps to 22.5% when using self-consistency. Hence, UL2 is able to leverage additional techniques that are unlocked by generating an intermediate reasoning chain.
>
> Thanks for asking us to clarify about other open-source models. We removed those claims.
>
> Thanks again for the feedback on the paper!

---

### Author Response · Authors · 2022-11-18
**Overall summary of rebuttal**

Hi folks,

Many thanks for taking time to review this paper and thanks for all the insights and feedback.

In general, we're really grateful for good scores.

Here's a short summary of the changes we made to the paper:
1. We fixed a typo of the Palm 8b result and modified the prose to reflect accordingly. We thank the reviewer who pointed this out.
2. We added a couple of qualitative examples to the Appendix of the paper. These are non-cherry picked examples. We are glad to include more samples if there is a need to.
3. We removed claims that this is the only open source model that leverage CoT.

We believe most of the legitimate concerns have been addressed.

Thanks for the feedback! :)

---

### Decision · Program_Chairs · 2023-01-20

**Decision:**

Accept: poster

**Justification For Why Not Higher Score:**

The does not contain innovative methods or ideas.

**Justification For Why Not Lower Score:**

The paper has important practical consequences and can help other researchers in their work.
The authors will release new pretrained models that perform well and will be useful to others.
Strong experimental and ablation studies support the main arguments.

**Metareview: Summary, Strengths And Weaknesses:**

The paper presents a unified perspective of different self-supervised pretraining objectives. More importantly, the paper shows how different objectives can be cast into each other and interpolated effectively. On the practical side, the paper proposes Mixture-of-Denoisers (MoD) pretraining and the notion of mode switching that allows models to be adapted to their downstream fine-tuning use. The paper shows experimentally that the proposed method can outperform T5 and/or GPT-like models across several different setups. The paper trains a 20B-parameter model that achieves SOTA performance on multiple well-established NLP tasks.

The paper does not contain novel creative ideas, but the experimental evaluation is very strong. The way that the space of pretraining methods is categorized and explored leads to new insights that, in terms, lead to a new state-of-the-art performance. The paper will likely impact how others think about pretraining objectives in the future.

**Note From Pc:**

if the above contains the word "oral" or "spotlight" please see: "oral" presentation means -> notable-top-5% and "spotlight" means -> notable-top-25%. As stated in our emails, we are disassociating presentation type from AC recommendations

**Summary Of Ac-Reviewer Meeting:**

AC review meeting was not held due to time constraints and low engagement from reviewers.